# Prefrontal gamma oscillations and fear extinction learning require early postnatal interneuron-oligodendroglia communication

Fabrice Plaisier [1,2], Cristóbal Ibaceta-González [2,7], Hasni Khelfaoui [2,7], Adrianne E. S. Watson[3], Corinne Poilbout [2,6], Anastassia Voronova [3,4] & María Cecilia Angulo [2,5] ✉

Emerging evidence links oligodendrocyte (OL) lineage cells and myelin to cognitive processes, yet the role of myelination in shaping neuronal networks critical for cognitive tasks remains unknown. We demonstrate in mice that early postnatal GABAergic signaling between interneurons and oligodendrocyte precursor cells (OPCs) is crucial for myelination of parvalbumin (PV) interneurons, which facilitates in vivo low-gamma oscillations in the medial prefrontal cortex (mPFC) and supports fear extinction learning. Disruption of this signaling results in PV interneuron dysmyelination, decreases low-gamma power, and impairs tone fear extinction. These deficits are specific to PV interneuron dysmyelination, as mPFC myelination, high-gamma oscillations and contextual fear extinction are not significantly altered. Increasing PV interneuron activity or enhancing myelination do not reverse the deficits, indicating the long-term consequences of these early myelination impairments. Our findings reveal the role of OPC GABAergic signaling in PV interneuron myelination and mPFC circuit maturation, with lasting impacts on gamma rhythms and cognition.

Oligodendrocytes (OLs) are specialized glial cells of the central nervous system (CNS) that produce myelin, a lipid-rich substance that envelops axons, facilitates faster impulse propagation and supplies energy. Beyond their cellular functions as myelinating glia, several lines of evidence highlight the pivotal role of OL lineage cells in the underpinnings of cognitive processes[1]. Behavioral interventions, such as fear conditioning tasks or working memory training in mice, enhance both oligodendrogenesis and myelination, while genetic models impeding OL differentiation result in compromised fear memory recall and spatial memory consolidation[2–4]. In humans, the correlation between white matter changes and cognitive functions further supports the link between OLs, myelin plasticity and cognition[1,5].

While cognitive processes rely on the orchestrated coordination of neuronal networks and the resulting brain oscillations, the contribution of glial cells to this orchestration remains underexplored.

[1]Inovarion SAS, Paris, France. [2]Université Paris Cité, Institute of Psychiatry and Neuroscience of Paris (IPNP), INSERM U1266, "Team: Interactions between neurons and oligodendroglia in myelination and myelin repair", Paris, France. [3]Department of Medical Genetics, Faculty of Medicine and Dentistry, University of Alberta, Edmonton, Alberta, Canada. [4]Department of Cell Biology, Faculty of Medicine and Dentistry, University of Alberta, Edmonton, Alberta, Canada. [5]GHU PARIS Psychiatrie & Neurosciences, Paris, France. [6]Present address: Inserm, UMR-S 1180, "Team: Signaling and cardiovascular pathophysiology", Université Paris-Saclay, Orsay, France. [7]These authors contributed equally: Cristóbal Ibaceta-González, Hasni Khelfaoui. ✉e-mail: maria-cecilia.angulo@u-paris.fr

Recent advances in mathematical modeling have provided insights into the potential role of myelin in promoting temporal synchronization among neuronal ensembles[6]. This theoretical framework is consistent with in vivo experiments conducted on the auditory system of dysmyelinated mouse models characterized by abnormal myelin formation, which exhibited delayed action potentials and loss of temporal precision[7]. However, the conversion of OL-induced neuronal synchronization into brain oscillation activities is still poorly understood. For instance, cuprizone demyelinated mice exhibit an enhanced rather than a decreased cortical theta power during phases of quiet wakefulness[8] and ablation of oligodendrogenesis affects in vivo electroencephalography (EEG) recordings even in the absence of task experience[9]. Moreover, besides myelin, the generation of new OLs themselves is crucial for coupling spindle oscillations in the prefrontal cortex with sharp wave ripple oscillations in the hippocampus following learning[2]. While these initial studies offer some insight, the influence of OL lineage cells on distinct oscillation frequencies that dictate cognitive performance remains elusive.

Cortical oscillations result from the dynamic interactions of excitatory and inhibitory neurons, with gamma oscillations being of particular importance for cognitive function[10]. Notably, gamma-band oscillations result from the fast synchronization of excitatory neuronal activity, a phenomenon largely modulated by parvalbumin (PV) interneurons that provide a robust synaptic inhibition[11–13]. Interestingly, PV interneurons share a unique association with OL lineage cells, arising from common embryonic origins[14]. During postnatal development, these interneurons form bona fide GABAergic synapses with OL precursor cells (OPCs) which disappear after the second postnatal week in mice[15–17]. Recent evidence highlights the critical role of early communication between PV interneurons and OPCs via GABA receptors in the myelination and maturation of PV interneurons, as well as in shaping cortical inhibitory circuit function and mouse behaviour[18,19]. Moreover, PV interneurons constitute the largest myelinated population of GABAergic interneurons in the cortex[20,21]. Considering the relationship between PV interneurons, OL lineage cells and myelin, we propose that early postnatal interactions between PV interneurons and OPCs play a crucial role in establishing prefrontal synchronization and oscillations in adulthood, especially in the gamma-frequency range, which could have significant implications for cognitive function.

To address our hypothesis, we genetically disrupted $GABA_A$ receptor-mediated synapses on OPCs during early postnatal development and subjected adult mice to a fear conditioning task while recording oscillations in the infralimbic (IL) region of the medial prefrontal cortex (mPFC), a key area for processing emotional and cognitive information[1]. Our results demonstrate that early postnatal disruption of OPC GABAergic synaptic signalling results in impairments in tone fear extinction learning which arise from a reduction in low-gamma frequency power during tone fear extinction sessions. These functional deficits are associated with PV interneuron dysmyelination and an excitation/inhibition (E/I) imbalance in the adult mPFC. Efforts to restore the altered phenotype in transgenic mice by local chemogenetic-mediated enhancement of PV interneuron activity in the mPFC or by in vivo infusion of the pro-myelinating compound fractalkine (FKN) fail to recover learning and memory deficits. Altogether, our findings reveal the role of early interneuron-OPC synaptic signalling in shaping mPFC circuits and associated fear extinction learning and memory, while also highlighting the long-lasting and irreversible impact of early OL-mediated disruptions on cognitive deficits.

## Results

### Early GABAergic OPC synapse disruption impairs tone fear extinction

To investigate the potential impact of early postnatal interneuron-OPC interactions on cognitive functioning in adulthood, we used the

$NG2cre^{ERT2+/-};Gcamp3^{f/f};\gamma2^{f/f}$ mice ($\gamma2^{f/f}$ mice) in which transient $\gamma2$ subunit-mediated GABAergic synapses of OPCs are inactivated by 4-hydroxytamoxifen (4-OHT) injections administered from postnatal day 3 (P3) to P5[22] (Supplementary Fig. 1a; see Online Methods). In agreement with our previous reports which demonstrated recombination at the $\gamma2$ locus and a partial reduction in OPC evoked GABAergic postsynaptic currents ($I_{GABA}$) in the somatosensory cortex[18,22], our current results reveal a near-complete abolishment of $I_{GABA}$ in layer 5 OPCs within the IL region of acute mPFC slices of $\gamma2^{f/f}$ mice at P9-P13 (Supplementary Fig. 1b, c). This finding indicates a more pronounced impairment of interneuron-OPC synaptic signaling in the mPFC compared to the somatosensory cortex.

To assess whether $\gamma2$-containing GABA-A receptor-mediated currents persist in OPCs into adulthood, we recorded $I_{GABA}$ in layer 5 OPCs of 2-month-old control and $\gamma2^{f/f}$ mice. In contrast to the pronounced $\gamma2$-dependent currents observed in young animals, evoked $I_{GABA}$ remained detectable at similar levels in both groups and was insensitive to zolpidem (1 μM), a positive allosteric modulator selective for GABA-A receptors containing the $\gamma2$ subunit (Supplementary Fig. 1a, d, e). This pharmacological profile as well as the presence of a GABAergic current in the $\gamma2^{f/f}$ mice is consistent with prior reports indicating that, while adult OPCs in the cortex and hippocampus express functional GABA-A receptors, these cells lack the $\gamma2$ subunit[23,24]. Together, these data confirm that $\gamma2$-mediated GABAergic synapses are developmentally restricted and absent in adult OPCs in the mPFC. To further validate the efficiency of OPC targeting in this region, we quantified recombination by immunolabeling Olig2 cells and enhancing GCaMP3 detection with an anti-GFP antibody in 4-month-old mice. We found 65.5% of recombinant OL lineage cells in control mice and 63.4% in $\gamma2^{f/f}$ mice in the mPFC, indicating efficient and comparable recombination for both groups (Supplementary Fig. 2a).

After confirming the early functional inactivation of OPC GABAergic synapses in this cortical region in $\gamma2^{f/f}$ mice, we next investigated whether this developmental synaptic alteration would affect mPFC-dependent cognitive functions in adulthood. To assess this, we first evaluated four-month-old $\gamma2^{f/f}$ mice using a fear conditioning task, a widely used paradigm to investigate mPFC-dependent processes such as learning, memory, emotional regulation, and perception[25].

To evaluate learning and memory processes associating environmental cues with aversive experiences, adult mice subjected to 4-OHT injections at P3-P5, underwent a behavioral test where a 30 s conditioned tone stimulus was paired with a foot shock delivered during the last 2 s of the tone in a specific context (Fig. 1a). In the conditioning session, adult control and $\gamma2^{f/f}$ mice exhibited similar fear memory acquisition, albeit slightly faster in $\gamma2^{f/f}$ mice (Fig. 1b). No significant differences were found between the two groups during the context extinction session (Fig. 1c). However, in the first tone extinction session, although the baseline level (BL, first trial block) of fear expression were comparable in control and $\gamma2^{f/f}$ mice (Fig. 1d), control mice showed faster and more extensive fear extinction learning by the end of the protocol (Fig. 1d). To test whether this deficit in fear extinction learning was persistent, we performed a second tone extinction session on the following day (extinction retrieval). We found that, in this session, control mice exhibited significantly lower baseline values and reduced initial freezing levels compared to the first tone extinction session, indicating successful extinction retrieval (Fig. 1e). The tone extinction learned in the first session was therefore retained and successfully retrieved over time in control mice. In contrast, $\gamma2^{f/f}$ mice showed persistently high baseline values and a weak extinction, with freezing levels remaining similar to those observed in the first tone extinction session (Fig. 1e), indicating that these mice failed to improve or consolidate fear extinction learning between sessions. Of note, no significant sex differences were observed between male and female mice in both control and $\gamma2^{f/f}$ groups (Supplementary

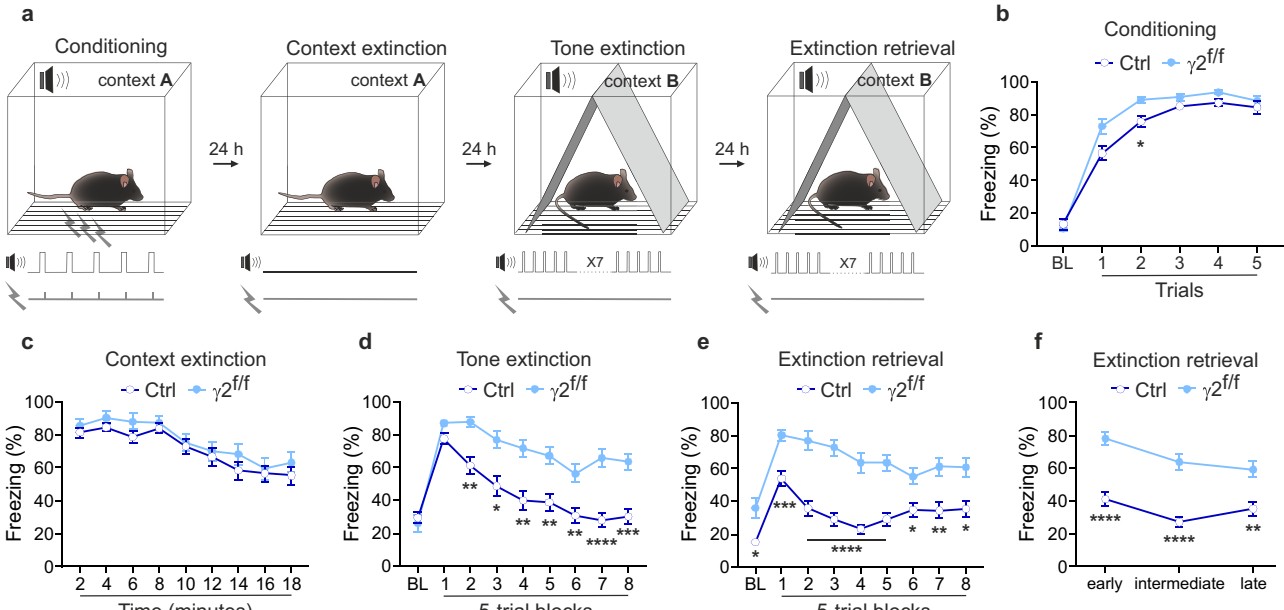

**Fig. 1 | Inactivation of GABAergic synapses in OPCs disrupts tone fear extinction. a** Mice were subjected to a four-day fear conditioning protocol consisting of four sessions: a conditioning session which included a 180 s baseline (BL) period followed by five 2-s foot shocks delivered at the end of each 30 s tone; a contextual extinction session (18 min); a tone extinction session which included a 180 s BL period followed by 40 tones of 30 s each with 5 s inter-trial interval (ITI); a tone extinction retrieval session which included a 180 s BL period followed by 40 tones of 30 s each with 5 s ITI. **b–e** Comparison of freezing behavior (percentage of time spent immobile) between control (dark blue) and γ2^f/f (light blue) mice during each session. No major differences were observed between the two groups during conditioning (**b**) and context extinction (**c**) sessions. Control mice: $N = 18$ and γ2^f/f mice: $N = 23$; $p = 0.9745/0.0748/0.0173/0.4112/0.1300/0.9752$ for BL and trials 1-5 in (**b**); $p = 0.9955/0.8627/0.7244/0.9989/>0.9999/>0.9999/0.9309/>0.9999/0.9705$ for 2–18 min in (**c**); two-way ANOVA test followed by post-hoc Sidak multiple comparisons test. Data are represented as mean±s.e.m. **d, e** During both tone extinction (**d**) and tone extinction retrieval (**e**) sessions, γ2^f/f mice exhibited significantly higher freezing levels across most 5-trial blocks compared to controls.

Note that in control mice, the BL and the two first freezing levels during extinction retrieval shown in (**e**) were significantly lower to BL and the two first freezing levels during tone extinction shown in (**d**) ($p = 0.0162$ for BL, $p = 0.0014$ for trial block 1 and $p = 0.007$ for trial block 2). Note also that BL freezing level was higher in γ2^f/f mice compared to controls during the extinction retrieval session (**e**). Control mice: $N = 18$ and γ2^f/f mice: $N = 23$; $p = 0.9998/0.1588/0.0013/0.0109/0.0017/0.0034/0.0078/<0.0001/0.0002$ for BL and 5-trial blocks 1-8 in (**d**); $p = 0.0358/0.0004/<0.0001/<0.0001/<0.0001/<0.0001/0.0229/0.0086/0.0136$ for BL and 5-trial blocks 1-8 in (**e**); two-way ANOVA test followed by post-hoc Sidak multiple comparisons test. Data are represented as mean±s.e.m. **f** Comparison of freezing behavior between control and γ2^f/f mice at early, intermediate and late stages of the extinction retrieval session (averaged freezing of 13 tones for each stage, see Supplementary Fig. 3i). Control mice: $N = 18$ and γ2^f/f mice: $N = 23$; $p < 0.0001/<0.0001/p = 0.0018$ at early-intermediate-late stages; two-way ANOVA test followed by post-hoc Sidak multiple comparisons test. Data are represented as mean ±s.e.m. *$p < 0.05$, **$p < 0.01$, ***$p < 0.001$, ****$p < 0.0001$. Source data are provided as a Source data file.

Fig. 3a–h). To confirm that the observed behavioral deficits specifically result from early postnatal recombination rather than to the presence of the floxed γ2 alleles, we reproduced the same fear conditioning paradigm in γ2 mice that were not injected with 4-OHT. These animals retain normal developmental expression of γ2 subunits in OPCs during development, while carrying the same Cre and γ2 alleles as the injected littermates. Non-injected γ2 mice showed no differences in fear conditioning behavior compared to controls (Supplementary Fig. 5a–e; Fig. 1b–f), indicating that the presence of the floxed alleles does not affect extinction behavior. We then assessed the effect of inducing recombination specifically in adulthood (Supplementary Fig. 5a). In this condition, no significant differences in tone extinction learning or retrieval were observed compared to non-injected γ2 mice (Supplementary Fig. 5b–e). These findings indicate that the behavioral deficits depend on early postnatal disruption of γ2-mediated GABAergic signaling in OPCs, and that recombination induction in adulthood, when the γ2 subunit is not expressed in OPCs, has no effect.

Given that the deficit was specific to tone extinction and not observed in context extinction, we considered whether this could be due to auditory impairments. To address this, we measured startle responses to noise stimuli of varying intensities, including the one used during fear conditioning (Supplementary Fig. 4a, b). Although this test does not assess potential hidden hearing loss[26], we found that both groups exhibited comparable startle responses to random noise of varying dB levels. This suggests that major auditory deficits are

unlikely to account for the observed differences in fear conditioning (Supplementary Fig. 4c, d). Building on these findings, we further analyzed the extinction learning deficit in γ2^f/f mice throughout different stages of the retrieval session (Supplementary Fig. 3i). The extinction learning deficit in γ2^f/f mice was predominant at early and intermediate stages, but it was also evident at the late stage (Fig. 1f), showing their impaired ability to acquire and consolidate an extinction memory.

Our results indicate that γ2^f/f mice exhibit reduced fear learning and memory capabilities, particularly in tone fear extinction learning, while no major auditory impairments were detected. These results suggest that the disruption of OPC GABAergic synaptic signaling during the early postnatal period leads to long-term cognitive impairments in adulthood, specifically affecting processes related to learning and memory, without evidence of substantial auditory deficits.

**Excitation-inhibition imbalance in the adult mPFC of γ2^f/f mice**

The long-lasting cognitive impairments observed in γ2^f/f mice, particularly during tone fear extinction, suggest that these deficits may be caused by underlying neurophysiological alterations. Therefore, we specifically tested the local excitation-inhibition (E/I) balance in the mPFC, a structure involved in regulating cognitive processes such as learning, memory, and emotional regulation[10,12]. This hypothesis is further supported by the known role of mPFC PV interneurons in

regulating the E/I balance by modulating excitatory inputs, which are necessary for higher-order cognitive functions[27]. Since fear conditioning experiments revealed specific deficits in tone fear extinction in γ2$^{f/f}$ mice, we focused on the IL region of the mPFC, a critical region for fear extinction memory processes[28].

To test the E/I balance in the IL region of adult γ2$^{f/f}$ mice, we performed patch-clamp recordings of layer 2/3 pyramidal cells in acute slices, while stimulating in layer V with an extracellular electrode. The stimulation elicited both excitatory (eEPSCs) and inhibitory (eIPSCs) postsynaptic currents in mature IL circuits (Fig. 2a). Our results revealed that, 3 months after inactivating the γ2 subunit-mediated GABAergic synapses in OPCs, γ2$^{f/f}$ mice exhibited a significant reduction in the mean amplitudes of eIPSCs, but not eEPSCs, in pyramidal neurons (Fig. 2a, b). Consequently, the E/I ratio was higher in γ2$^{f/f}$ mice, causing a strong E/I imbalance (Fig. 2b).

To assess whether the E/I imbalance observed in the mPFC reflects a broader effect or is more regionally restricted, we extended our analysis to two additional brain regions involved in fear expression and extinction: the amygdala and hippocampus. First, we recorded from principal neurons in the basolateral amygdala (BLA), while stimulating the lateral amygdala (Fig. 2c, d). Similarly, we recorded from CA1 pyramidal neurons in the hippocampus, while stimulating Schaffer collateral inputs from CA3 (Fig. 2e, f). Unlike the IL region, we found no significant differences in eIPSC and eEPSC amplitudes, nor in the E/I ratio between control and γ2$^{f/f}$ mice in either the BLA or CA1 regions (Fig. 2d, f).

To further evaluate the functional specificity of these alterations, we complemented these recordings with region-specific behavioral tests targeting the amygdala and the hippocampus: the elevated plus maze and the novel object recognition, respectively. In both tasks, γ2$^{f/f}$ mice performed similarly to controls, showing no significant differences in anxiety levels and recognition memory (Supplementary Fig. 5f, g). These results are consistent with the lack of synaptic alterations in these regions. Moreover, they align with our fear conditioning results, in which γ2$^{f/f}$ mice exhibited normal fear acquisition, a process largely mediated by the amygdala, and normal contextual extinction, which strongly depends on hippocampal function (Figs. 1b, c).

Since the proper regulation of E/I balance is essential for normal cognitive processing, the observed alterations in the mPFC functioning suggest a functional impairment in this region due to early interneuron-OPC synapse disruption. These findings indicate that the synaptic E/I imbalance is more prominent in the mPFC than the amygdala and hippocampus, suggesting that the mPFC is particularly sensitive to early developmental perturbations.

## PV interneuron dysmyelination in the adult mPFC of γ2$^{f/f}$ mice

The observed cognitive and E/I ratio deficits prompted us to investigate whether early disruption of interneuron-OPC synapses results in persistent myelination deficits, particularly in PV interneurons known for their highly plastic myelination[29]. Previous findings in the somatosensory cortex demonstrated that γ2$^{f/f}$ mice exhibit myelination defects in PV interneurons during juvenile stages[18]. Here, we extended this analysis to the mPFC in adulthood to determine whether these abnormalities persist and specifically affect the myelination of PV interneurons, the primary source of transient GABAergic synaptic input onto OPCs[16,17]. We found that γ2$^{f/f}$ mice displayed normal overall myelin coverage and intensity of MBP staining in IL cortical layers (Fig. 3a–c). We then analyzed the length of individual myelin segments, or internodes, along PV$^+$ axons in the same region. This was achieved by reconstructing high resolution images of axonal segments expressing MBP, PV and the axonal marker SMI-312 (Fig. 3d). Measurements revealed a statistically significant increase in the mean internode length of PV$^+$, but not PV$^-$, axons of γ2$^{f/f}$ mice compared to controls (Fig. 3e, f). The myelination defect observed in PV$^+$ axons was not

accompanied by alterations in either the cell density or distribution of PV interneurons across mPFC layers, nor by changes in the total interneuron population identified by GAD67 expression (Supplementary Fig. 6a–e). In addition, quantification of PV signal within SMI-312$^+$ axonal compartments–used as a proxy for PV$^+$ axon density–did not reveal significant differences between control and γ2 mice (Supplementary Fig. 6f, g). Furthermore, we found no differences in the cell density or proportion of Olig2$^+$/CC1$^-$ OPCs and Olig2$^+$/CC1$^+$ OLs between control and γ2$^{f/f}$ mice (Supplementary Fig. 2b–d). These results suggest that the early disruption of γ2-mediated synapses in OPCs impacts predominantly PV interneuron myelination, without affecting interneuron survival or the maturation of OL lineage cells in the mPFC.

Although OPCs form GABAergic synapses only transiently during postnatal development and these synapses are absent in adulthood[15,16,23], myelination defects in PV interneurons persists into adulthood of γ2$^{f/f}$ mice. Notably, these alterations are specific to PV interneurons as we found no significant changes in the overall mPFC myelination pattern, length of internodes of PV$^-$ axons, OL lineage cell densities, PV interneuron survival, the distribution of interneurons within cortical layers or of the estimated PV$^+$ axon density.

## Early GABAergic OPC synapse loss irreversibly impairs fear extinction

The persistent PV interneuron dysmyelination as well as the electrophysiological and cognitive impairments observed in γ2$^{f/f}$ mice raised the question of whether these deficits could be corrected through targeted interventions. To this end, we explored two distinct strategies aimed at addressing the underlying structural and behavioral abnormalities: enhancing PV interneuron activity and promoting myelination.

Previous research has demonstrated that using a modified form of the human M3 muscarinic (hM3Dq) receptor, activated by injections of the inert clozapine metabolite clozapine-N-oxide (CNO) for 14 days, can specifically promote activity-dependent myelination of PV interneurons in the adult mPFC[30]. In addition, acutely increasing PV interneuron activity with a chemogenetic approach should also rebalance the observed reduced cortical inhibition, improving E/I impairments (Fig. 2a, b). Building on this finding, we explored whether selectively targeting PV interneurons with a chemogenetic approach could effectively address dysmyelination and fear learning deficits (Fig. 4a). We performed bilateral intracranial injections into the IL region of the mPFC in γ2$^{f/f}$ mice using either pAAV-S5E2-dTom-nlsdTom (dTom) or pAAV-S5E2-hM3D(Gq)-P2A-dTomato (hM3Dq-dTom) viral vectors[31] (Fig. 4a, b). Approximately 80% of cells infected by these viral constructs were identified as PV$^+$ interneurons in this region (Fig. 4b, c). To confirm the functional efficacy of the hM3Dq-mediated activation of PV interneurons, we first performed whole-cell patch-clamp recordings in acute mPFC slices from γ2$^{f/f}$ mice injected with pAAV-S5E2-hM3D(Gq)-P2A-dTomato. The dTom$^+$ neurons, that express hM3Dq, exhibited an increased excitability in the presence of CNO while nearby dTom$^-$ pyramidal neurons did not, indicating that CNO effectively enhanced PV interneuron activity (Supplementary Fig. 7a–d). This supports the specificity and efficacy of hM3Dq-mediated chemogenetic manipulation. We then performed twice-daily in vivo injections of CNO over two weeks in dTom-injected and hM3Dq-dTom-injected γ2$^{f/f}$ mice (Fig. 4a). Our results demonstrate that CNO administration did not induce any significant changes in either the coverage or mean intensity of the MBP fluorescence signal compared to those in dTom-injected γ2$^{f/f}$ mice (Fig. 4d-f). Furthermore, we did not detect any reduction of PV interneuron internode length in hM3Dq-dTom-injected γ2$^{f/f}$ mice (Fig. 4g), suggesting that increased PV interneuron activity in adulthood does not restore aberrant internode length. To examine whether enhancing PV interneuron activity in vivo, independently of the effect on interneuron dysmyelination, ameliorates

 

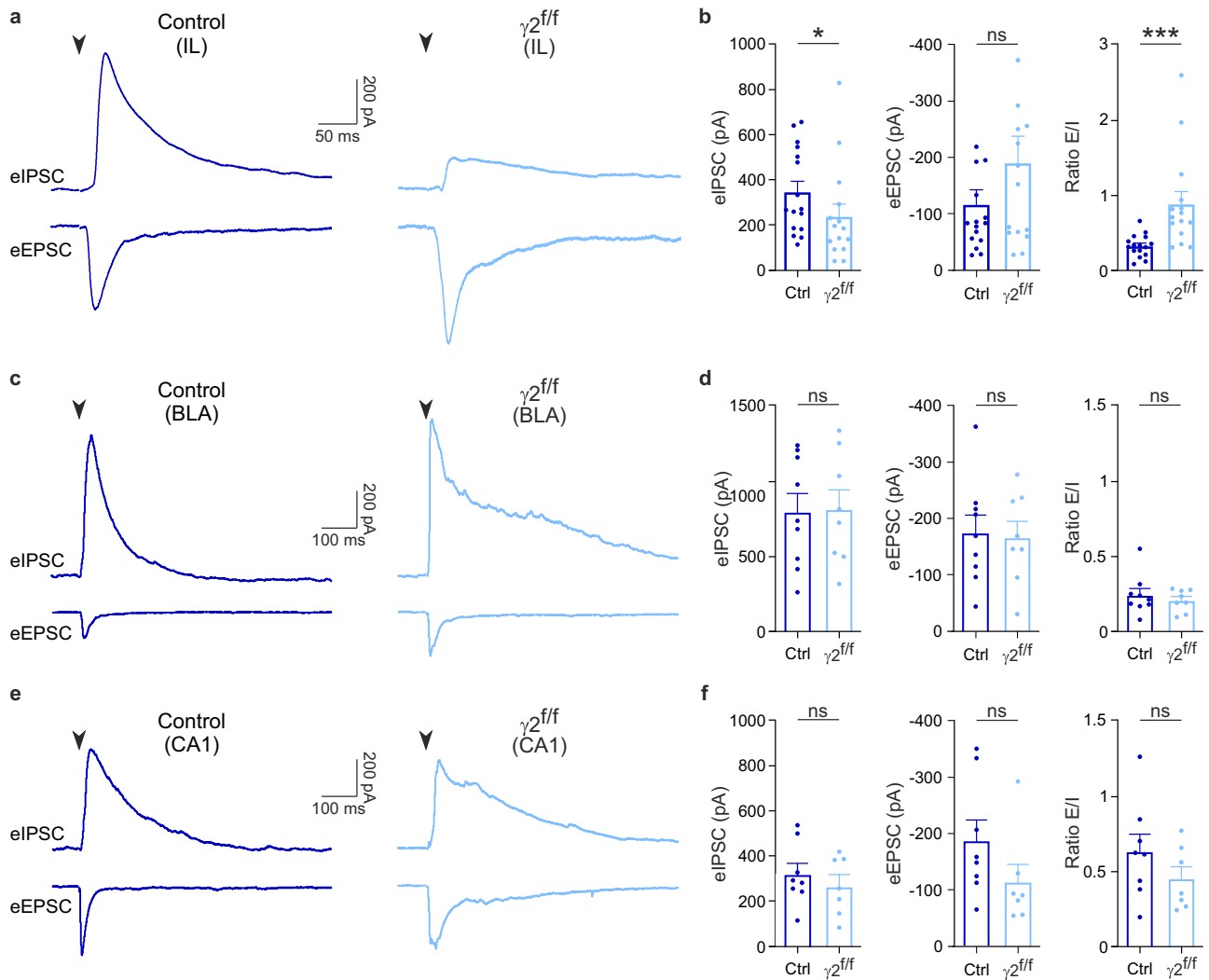

**Fig. 2 | Early inactivation of GABAergic synapses of OPCs impairs inhibition in the adult mPFC. a** eIPSCs (top trace) and eEPSCs (bottom trace) in layer 2/3 pyramidal neurons held at 0 mV and −70 mV, respectively, upon extracellular stimulation of layer 5 of the infralimbic (IL) region of acute slices of the mPFC (*inset*, left) in 3-month-old control (dark blue) and γ2^{f/f} (light blue) mice. Stimulation artifacts blanked, stimulation time indicated (black arrowheads). **b** Dot plots of eIPSCs (left), eEPSCs (middle) and E/I ratio (right) calculated from eEPSCs and eIPSCs of layer 2/3 pyramidal neurons evoked by extracellular stimulation in layer 5 in control (dark blue) and γ2^{f/f} mice (light blue) (n = 16 cells from N = 9 mice and n = 15 cells from N = 7 mice for control and γ2^{f/f} mice, respectively). p = 0.0494 for eIPSCs, p = 0.3378 for eEPSCs, p = 0.0002 for E/I ratio; two-tailed Mann–Whitney U test. Data are represented as mean±s.e.m. *p < 0.05, ***p < 0.001, ns: not significant. **c** eIPSCs (top trace) and eEPSCs (bottom trace) in principal neurons of the basolateral amygdala (BLA) held at 0 mV and −70 mV, respectively, upon extracellular stimulation of the lateral amygdala (LA) in 4-month-old control (dark blue) and γ2^{f/f} (light blue) mice. Stimulation artifacts blanked, stimulation time indicated (black arrowheads). **d** Dot plots of eIPSCs (left) and eEPSCs (middle) and E/I ratio (right) calculated from

eEPSCs and eIPSCs of principal neurons evoked by extracellular stimulation in the LA in control (dark blue) and γ2^{f/f} mice (light blue) (n = 9 cells from N = 5 mice and n = 8 cells from N = 3 mice for control and γ2^{f/f} mice, respectively). p = 0.9082 for eIPSCs, p = 0.8365 for eEPSCs, p = 0.4968 for E/I ratio; two-tailed Student t test. Data are represented as mean±s.e.m; ns: not significant. **e** eIPSCs (top trace) and eEPSCs (bottom trace) in CA1 pyramidal neurons of the hippocampus held at 0 mV and −70 mV, respectively, upon extracellular stimulation of Schaffer collaterals in 4-month-old control (dark blue) and γ2^{f/f} (light blue) mice. Stimulation artifacts blanked, stimulation time indicated (black arrowheads). **f** Dot plots of eIPSCs (left), eEPSCs (middle) and E/I ratio (right) calculated from eEPSCs and eIPSCs of pyramidal neurons evoked by extracellular stimulation in stratum radiatum in control (dark blue) and γ2^{f/f} mice (light blue) (n = 8 cells from 5 mice and n = 7 cells from 4 mice for control and γ2^{f/f} mice, respectively). p = 0.4531 for eIPSCs, p = 0.0721 for eEPSCs, p = 0.2382 for E/I ratio; two-tailed Student t test for eIPSCs and E/I ratio; two-tailed Mann–Whitney U test for eEPSCs. Data are represented as mean±s.e.m. ns: not significant. Source data are provided as a Source data file.

cognitive deficits, we conducted the fear conditioning task in both CNO-treated controls and γ2^{f/f} mice. As expected, no differences were observed in the acquisition or context sessions (Supplementary Fig. 8a, b). Consistent with the lack of effect on PV interneuron internode length, we observed no significant differences during the tone extinction and extinction retrieval sessions between the two viral injected groups (Fig. 4h, i), nor when compared to naive γ2^{f/f} mice (Fig. 1d, e).

Our second strategy consisted in enhancing myelination in young adult mice, a developmental period when myelination is still actively

progressing in the mPFC and cognitive deficits associated with neurodevelopmental disorders are often recognized due to the emergence of visible symptoms[1]. We reasoned that targeting this critical period might offer greater potential for modifying myelination and improving cognitive deficits because: (1) myelination plasticity persists well beyond early postnatal stages in cortical regions, and (2) therapeutic interventions during active myelination phases might rescue structural and functional deficits caused by earlier disruptions. We focused on fractalkine (FKN), a well-studied chemokine known for its role in microglial function[32], but also shown to promote OPC

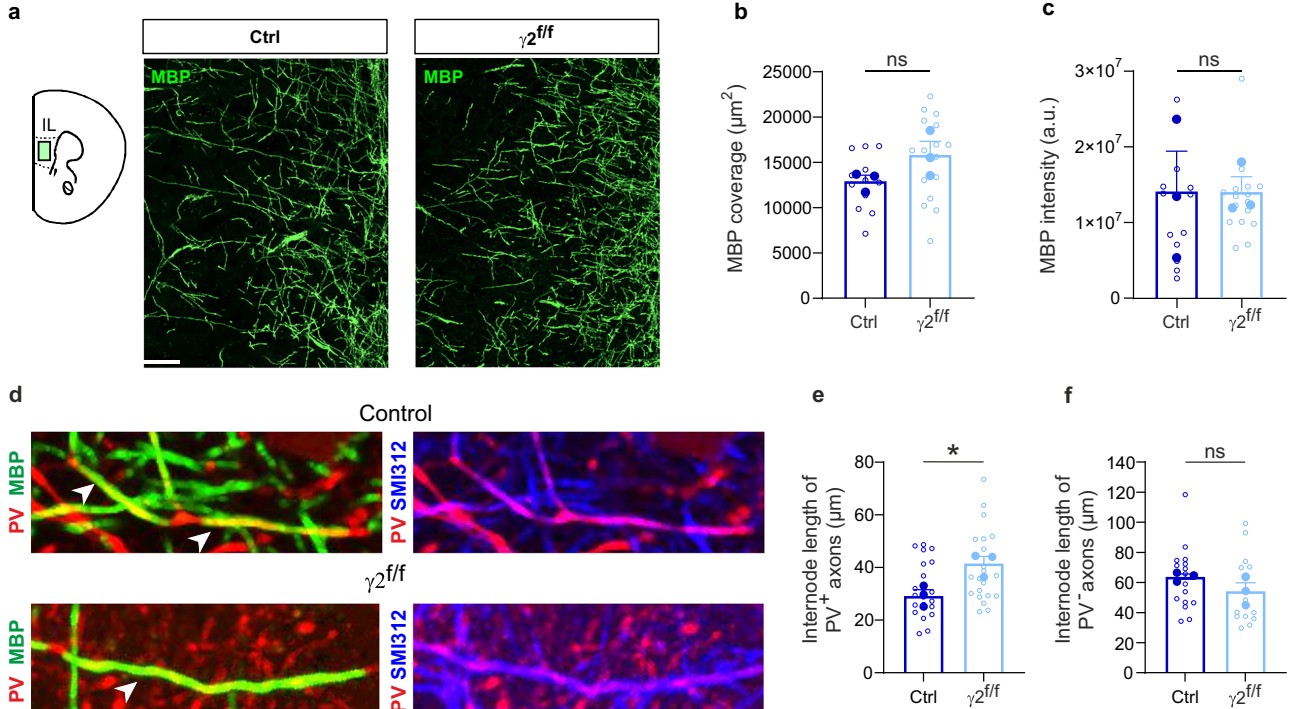

**Fig. 3 | Inactivation of GABAergic synapses of OPCs impairs PV interneuron internode length in the mPFC. a** Confocal images of MBP in the IL region of the mPFC of control (left) and γ2^f/f (right) mice. Scale bar: 50 μm. **b, c** Dot plots of MBP fluorescence coverage (**b**) and intensity (**c**) of control (dark blue) and γ2^f/f (light blue) mice. Control mice: *n* = 12 technical replicates from *N* = 3 mice (biological replicates) and γ2^f/f mice: *n* = 16 technical replicates from *N* = 3 mice (biological replicates). *p* = 0.078 for (**b**) and *p* = 0.374 for (**c**); two-tailed linear mixed models with likelihood ratio tests applied to account for repeated measures (open dots) within each animal (closed dots). Data are presented as mean±s.e.m. ns: not significant. **d** Confocal images of PV (red), MBP (green), and SMI-312 (blue) labeling, illustrating myelinated internodes of PV^+ axons, identified by bare axonal regions on each side, in control and γ2^f/f mice (white arrowheads). Note that the internode

of PV^+ axons in γ2^f/f mice appeared longer compared to those of control mice. Scale bar: 10 μm. **e, f** Dot plots of internode lengths of PV^+ (**e**) and PV^- (**f**) axons in control (dark blue) and γ2^f/f (light blue) mice. Control mice: *n* = 21 technical replicates from *N* = 3 mice (biological replicates) and γ2^f/f mice: *n* = 21 technical replicates from *N* = 3 mice (biological replicates) for PV^+ axons; Control mice: *n* = 16 technical replicates from *N* = 3 mice (biological replicates) and γ2^f/f mice: *n* = 15 technical replicates from *N* = 3 mice (biological replicates) for PV^- axons. *p* = 0.026 for (**e**) and *p* = 0.325 for (**f**); two-tailed linear mixed models with likelihood ratio tests applied to account for repeated measures (open dots) within each animal (closed dots). Data are represented as mean±s.e.m. *p < 0.05, ns: not significant. Source data are provided as a Source data file.

differentiation into mature OLs during cortical development through its release from GABAergic interneurons[33]. Recent studies demonstrate that FKN can promote myelination under both healthy and demyelinating conditions when administered exogenously[34,35]. We thus exploited the pro-myelinating potential of FKN to examine whether its exogenous application in mice could compensate for PV interneuron dysmyelination in the mPFC and mitigate the associated learning and memory deficits observed during tone fear extinction. To this end, we performed in vivo FKN infusions into the lateral ventricles of P42-46 mice over a 25–28-day period (Fig. 5a). Our results revealed that in vivo FKN infusions in γ2^f/f mice increased both the coverage and mean intensity of MBP fluorescent signal in the mPFC of infused mice compared to vehicle-treated mice, indicating an enhanced myelination (Fig. 5b–d). However, measurements of internode length of PV^+/SMI-312^+ axons in the same animals showed no difference between the two groups, suggesting that this intervention is insufficient to repair PV interneuron-specific myelin abnormalities in γ2^f/f mice (Fig. 5e). To assess whether FKN-induced myelination enhancement could still result in cognitive improvements, we conducted the fear conditioning task in γ2^f/f mice treated with either vehicle or FKN for comparison. As expected, no differences were observed in the acquisition or context sessions (Supplementary Fig. 8c, d). However, no improvements were observed in the tone extinction or extinction retrieval sessions either (Fig. 5f, g), indicating that enhancing ongoing mPFC myelination fails to rectify PV interneuron dysmyelination or improve the associated learning and memory deficits.

Our findings show the irreversible cognitive impairments resulting from early disruption of OPC GABAergic synapses. Despite targeted interventions aimed at increasing PV interneuron activity or enhancing myelination, we observed no improvements in PV interneuron internode length and fear extinction learning and memory. Once early interneuron-OPC interactions are compromised during a critical developmental window, subsequent efforts to reverse structural and functional deficits become particularly challenging. The early postnatal period, particularly the first two postnatal weeks[16], is therefore essential for ensuring proper cognitive development and function.

**Specific alterations in gamma oscillations during fear extinction**

The E/I balance in the mPFC, which is disrupted in γ2^f/f mice, plays a fundamental role in generating gamma oscillations (30–90 Hz), high-frequency neural rhythms which support prefrontal-dependent cognitive processing. These oscillations arise from the rapid synchronization of excitatory activity, with PV interneurons playing a central role by providing perisomatic inhibition onto pyramidal cell ensembles, which fine-tunes network discharges[10,12,13]. Deficits in E/I balance and PV interneuron myelination could potentially disrupt gamma oscillations, impacting cognitive functions. To investigate this possibility, we recorded local field potentials (LFPs) from the IL region of the mPFC in freely moving mice using an implanted tetrode (Fig. 6a), of which the placement was confirmed postmortem through histological analysis (Fig. 6b). After 10 days of recovery post-implantation, mice were first

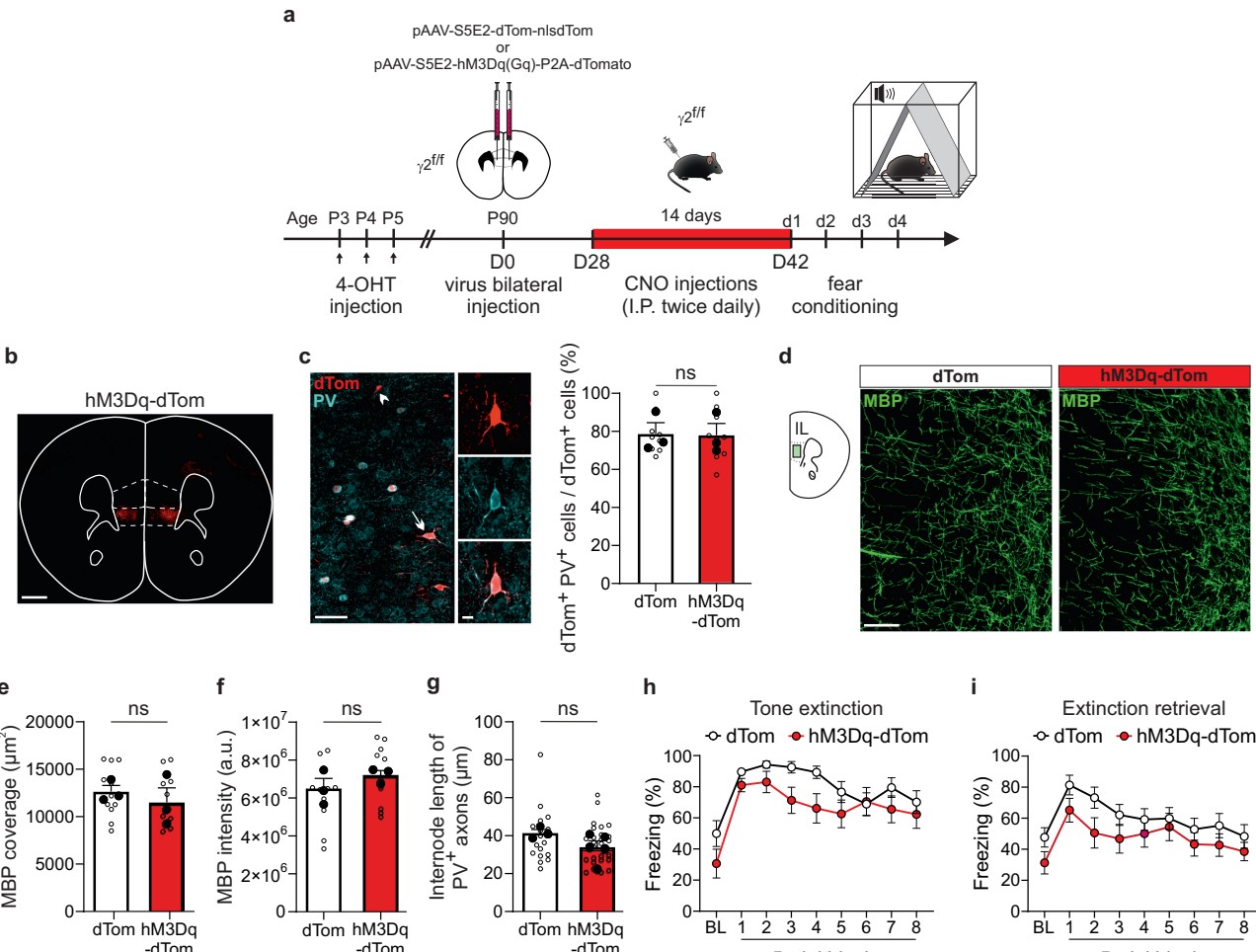

**Fig. 4 | Chemogenetic-induced increase in PV interneuron activity does not rescue fear extinction deficits. a** Schematic of viral injections and clozapine-N-oxide (CNO) administration protocol for PV interneuron activation in the mPFC. The $\gamma2^{f/f}$ mice received bilateral injections of either pAAV-S5E2-dTom-nlsd (dTom) or pAAV-S5E2-hM3Dq(Gq)-P2A-dTomato (hM3Dq-dTom) virus targeting PV interneurons in the mPFC at around P90. After one month, the same mice received twice-daily intraperitoneal injections of CNO for 14 consecutive days. The day following the final CNO injection, they were subjected to the fear conditioning protocol (d1–d4). **b** Confocal image of a coronal section showing the restricted expression of hM3Dq-dTom virus on the IL region of the mPFC. Scale bar: 500 μm. **c** Confocal images of dTom⁺/PV⁺ neurons in the IL region. A dTom⁺/PV⁺ cell is shown (Inset, arrow); a dTom⁺/PV⁻ cells is indicated (arrowhead). Dot plots (right) of the percentage of dTom⁺/PV⁺ cells in both dTom-injected (white) and hM3Dq-dTom-injected (red) $\gamma2^{f/f}$ mice. dTom-injected $\gamma2^{f/f}$ mice: $n = 9$ technical replicates from $N = 3$ mice (biological replicates) and hM3Dq-dTom-injected $\gamma2^{f/f}$ mice: $n = 9$ technical replicates from $N = 3$ mice (biological replicates). $p = 0.92$ for (**c**); two-tailed linear mixed models with likelihood ratio tests applied to account for repeated measures (open dots) within each animal (closed dots). Data are represented as mean±s.e.m. ns: not significant. Scale bar: 50 μm and 10 μm. **d** Confocal images of MBP in the IL region of the mPFC of dTom-injected (left) and hM3Dq-dTom-injected (right) $\gamma2^{f/f}$ mice. Scale bar: 50 μm. **e, f** Dot plots of MBP fluorescence coverage (**e**) and intensity (**f**) of the IL region of dTom-injected (white) and hM3Dq-

dTom-injected (red) $\gamma2^{f/f}$ mice (dTom-injected $\gamma2^{f/f}$ mice: $n = 11$ technical replicates from $N = 3$ mice (biological replicates) and hM3Dq-dTom-injected $\gamma2^{f/f}$ mice: $n = 10$ technical replicates from $N = 3$ mice (biological replicates)). $p = 0.486$ for (**e**), $p = 0.158$ for (**f**); two-tailed linear mixed models with likelihood ratio tests applied to account for repeated measures (open dots) within each animal (closed dots). Data are presented as mean±s.e.m. ns: not significant. **g** Dot plots of internode lengths of PV⁺ axons in dTom-injected (white) and hM3Dq-dTom-injected (red) $\gamma2^{f/f}$ mice (dTom-injected $\gamma2^{f/f}$ mice: $n = 19$ technical replicates from $N = 3$ mice (biological replicates) and hM3Dq-dTom-injected $\gamma2^{f/f}$ mice: $n = 26$ technical replicates from $N = 5$ mice (biological replicates). $p = 0.181$ for (**g**); two-tailed linear mixed models with likelihood ratio tests applied to account for repeated measures (open dots) within each animal (closed dots). Data are represented as mean±s.e.m. ns: not significant. **h, i** Comparison of freezing behavior between dTom-injected (white) and hM3Dq-dTom-injected (red) $\gamma2^{f/f}$ mice during tone extinction (**h**) and tone extinction retrieval (**i**) sessions. No significant differences were observed between the two groups (dTom-injected $\gamma2^{f/f}$ mice: $N = 13$ and hM3Dq-dTom-injected $\gamma2^{f/f}$ mice: $N = 14$). $p = 0.7225/0.5369/0.7494/0.2596/0.2849/0.8932/>0.9999/0.8701/$ 0.9983 for BL and 5-trial blocks 1-8 in (**h**) and $p = 0.5947/0.6679/0.4744/0.8512/$ 0.9922/0.9999/0.9819/0.9242/0.9711 for BL and 5-trial blocks 1-8 in (**i**); two-way ANOVA test followed by post-hoc Sidak multiple comparisons test. Data are represented as mean±s.e.m. ns: not significant. 4-OHT: 4-hydroxytamoxifen. Source data are provided as a Source data file.

habituated to wearing the recording cable in their home cage for 3 days. Then, they were placed in an open field environment and later subjected to the fear conditioning task, during which LFP signals were recorded (Fig. 6a, c). We analyzed these signals using fast Fourier transformation (FFT) to generate spectrograms and power spectra allowing us to extract different oscillatory activities during the behavioral tasks (Fig. 6c–f, i, j).

In vivo LFP recordings during the open field test detected both theta (4–12 Hz) and gamma (30–90 Hz) oscillations in the mPFC of control and $\gamma2^{f/f}$ mice (Fig. 6c). Contrary to our hypothesis that disrupted myelination would impair gamma oscillations, there were no significant differences in theta or gamma oscillatory activity between the two groups, regardless of whether the mice were in the center or periphery of the arena (Fig. 6e–h). These results are

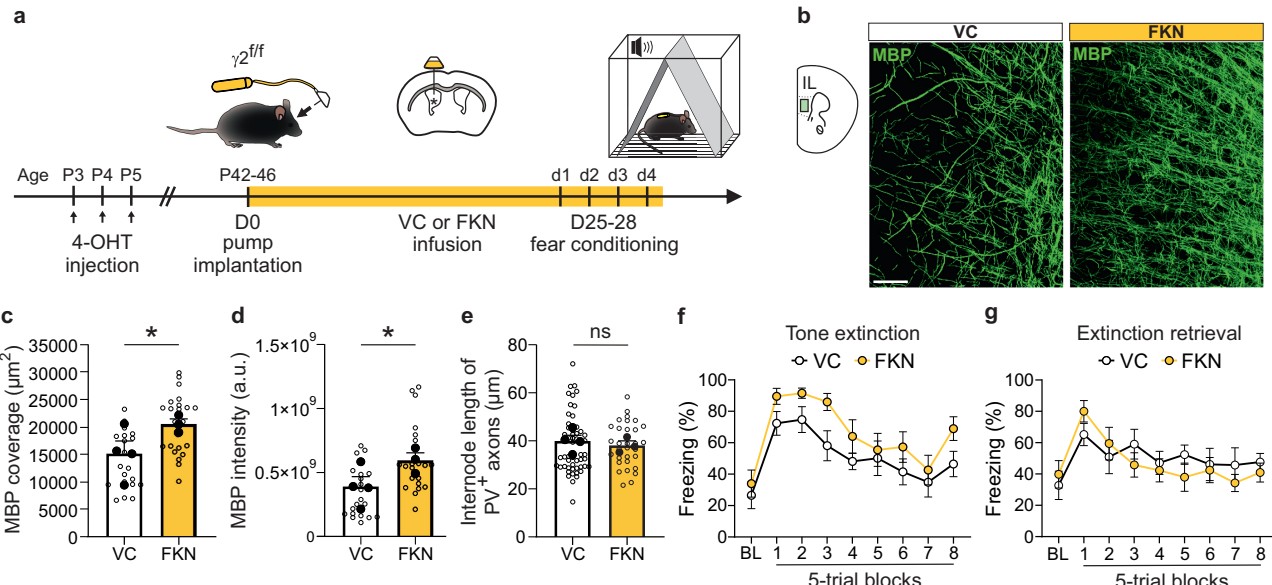

**Fig. 5 | In vivo FKN infusions increase myelination, but does not rescue fear extinction deficits. a** Schematic of the in vivo infusion protocol. Vehicle (VC) or FKN was continuously infused into the lateral ventricle of γ2$^{f/f}$ mice at P42-P46 for 25–28 days (D25-D28) using a micro-osmotic pump. These mice were subjected to the fear conditioning protocol (d1–d4) during the last days of infusion. **b** Confocal images of MBP in the IL region of the mPFC of VC-infused (left) and FKN-infused (right) γ2$^{f/f}$ mice at P70-74. Note the increased myelination of FKN-infused γ2$^{f/f}$ mouse. Scale bar: 50 μm. **c, d** Dot plots of MBP fluorescence coverage (**c**) and intensity (**d**) of the IL region of VC-infused (white) and FKN-infused (yellow) γ2$^{f/f}$ mice. VC-infused mice: $n = 21$ technical replicates from $N = 4$ mice (biological replicates) and FKN-infused mice: $n = 23$ technical replicates from $N = 3$ mice (biological replicates); $p = 0.021$ for (**c**) and $p = 0.016$ for (**d**); two-tailed linear mixed models with likelihood ratio tests applied to account for repeated measures (open dots) within each animal (closed dots). Data are presented as mean±s.e.m. *$p < 0.05$. **e** Dot plots of internode lengths of PV$^+$ axons in VC-infused (white) and FKN-infused

(yellow) γ2$^{f/f}$ mice. VC-infused mice: $n = 56$ technical replicates from $N = 4$ mice (biological replicates) and FKN-infused mice: $n = 31$ technical replicates from $N = 3$ mice (biological replicates); $p = 0.722$ for (**e**); two-tailed linear mixed models with likelihood ratio tests applied to account for repeated measures (open dots) within each animal (closed dots). Data are represented as mean±s.e.m. ns: not significant. **f, g** Comparison of freezing behavior between VC-infused (white) and FKN-infused (yellow) γ2$^{f/f}$ mice during tone extinction (**f**) and tone extinction retrieval (**g**) sessions. No significant differences were observed between the two groups. VC-infused mice: $N = 8$ and FKN-infused mice: $N = 8$; $p = 0.9992/0.5402/0.5641/0.2220/0.8762/>0.9999/0.9109/0.9995/0.4330$ for BL and 5-trial blocks 1-8 in (**f**); $p = 0.9996/0.7776/0.9995/0.9590/>0.9999/0.8786/>0.9999/0.9845/0.9933$ for BL and 5-trial blocks 1-8 in (**g**); two-way ANOVA test followed by post-hoc Sidak multiple comparisons test. Data are represented as mean±s.e.m. Source data are provided as a Source data file.

consistent with the absence of significant differences in exploratory behavior between control and γ2$^{f/f}$ mice, including distance traveled, time spent, immobility time in the center and periphery of the arena, as well as the number of crossings or average speed (Supplementary Fig. 9a–f). These findings indicate that deficits in PV interneuron myelination did not result in alterations in either oscillatory activity of IL region or exploratory behavior under the conditions of the open field task.

The consequences of PV interneuron dysmyelination in the mPFC may be task-specific, potentially impacting complex processes rather than general locomotor or exploratory behaviors. These same implanted mice were therefore subjected to the fear conditioning protocol which is designed to assess more cognitively demanding processes (Fig. 6a). In vivo LFP recordings were performed during the tone extinction retrieval session, when more pronounced behavioral differences between control and γ2$^{f/f}$ mice were observed (Fig. 1e). During this session, we optimized recordings by using a Faraday cage and minimizing external noise. Analysis of the oscillatory activity revealed no significant differences in the relative power of theta oscillations between the two groups during tone presentation and inter-trial intervals (ITI) (Fig. 6i–k). However, γ2$^{f/f}$ mice exhibited a significant reduction in gamma oscillations throughout both periods, suggesting a specific disruption of neural dynamics at this frequency range (Fig. 6i, j, l). These results indicate that early postnatal disruption of interneuron-OPC signaling leads to persistent PV interneuron dysmyelination and E/I imbalance in the mPFC, which in turn drive alterations in gamma oscillations and the subsequent impairments in fear extinction learning observed in γ2$^{f/f}$ mice.

The absence of significant gamma oscillation changes in the open field, in contrast to the pronounced reduction observed during fear extinction retrieval, suggests that PV interneuron dysmyelination predominantly affects higher cognitive processes, without necessarily disrupting basal neural activity or basic behaviors like exploration or locomotion.

## Low-gamma oscillation disruption during fear extinction retrieval

Gamma oscillations can be subdivided into low and high frequency ranges, each contributing distinctly to cognitive processes. High-gamma frequencies have been associated with encoding processes, whereas low-gamma frequencies support retrieval in human episodic memory[36]. To investigate whether the differences in gamma oscillations observed during fear extinction retrieval between control and γ2$^{f/f}$ mice were frequency-specific, we conducted a detailed analysis of both low-gamma (30–45 Hz) and high-gamma (55–90 Hz) power. Our results revealed a significant reduction in the relative power of low-gamma oscillations in γ2$^{f/f}$ mice during both tone presentation and ITI of fear extinction retrieval, while high-gamma oscillations showed no significant differences (Fig. 7a, b). These results suggest that deficits in neuronal processing in mPFC of γ2$^{f/f}$ mice are primarily associated with alterations in low-gamma oscillations.

Gamma oscillations are known to couple strongly with theta oscillations during cognitive processes, playing a key role in coordinating network-wide communication[37]. Our analyses reveal that the strength of the theta phase-low-gamma amplitude and theta phase-high-gamma amplitude coupling increased during tone presentation

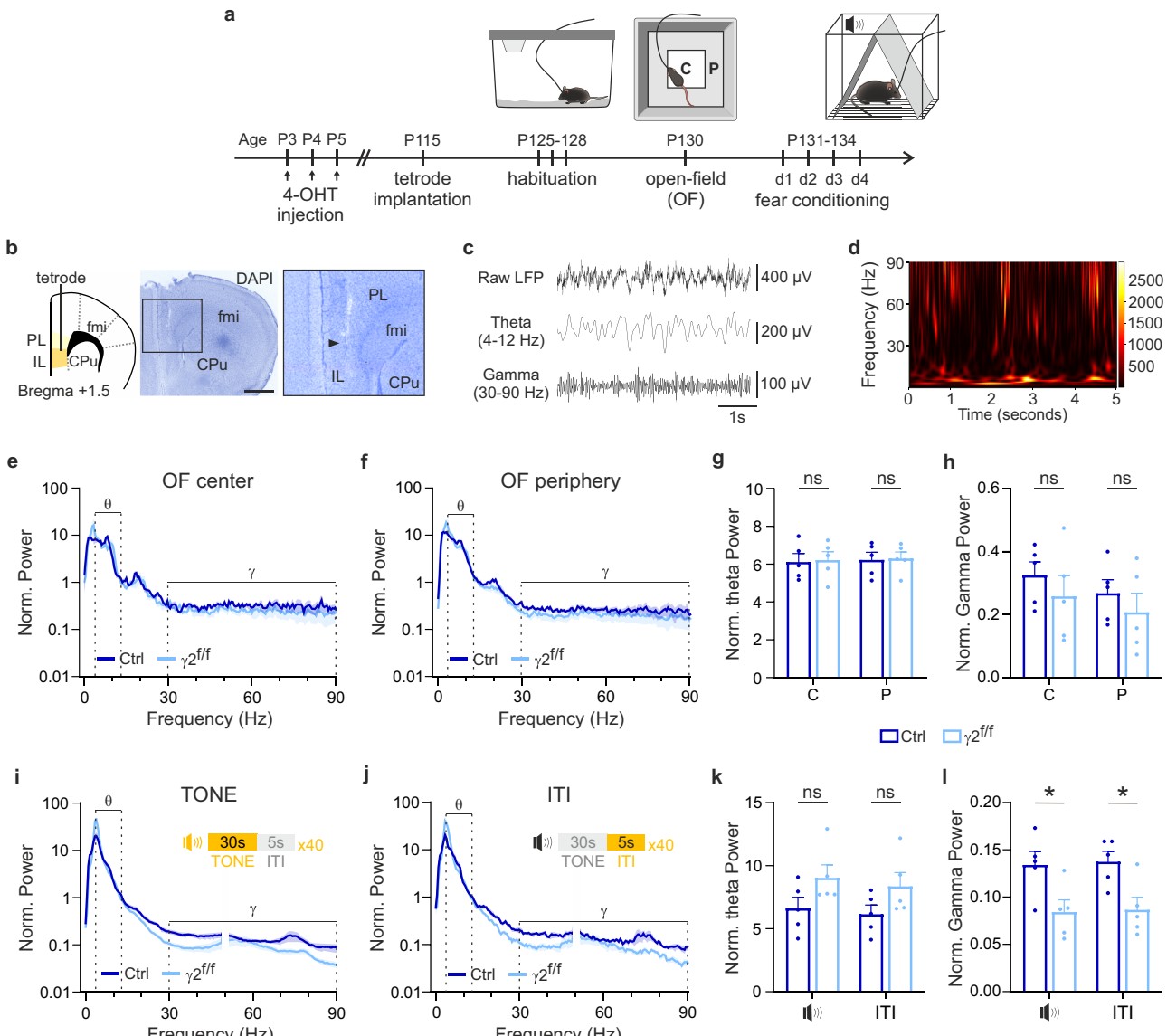

**Fig. 6 | Disruption of gamma oscillations during fear extinction retrieval, but not exploration. a** Schematic of the protocol for conducting in vivo recordings in behaving mice. Surgery to implant the microdrive containing the tetrode was performed at around P115. Recordings of LFPs were performed 7–15 days post-surgery during open-field exploration and fear extinction retrieval. **b** The tetrode was implanted in the IL region of the mPFC (left). Brain slice of a recorded mouse (middle) after a DAPI staining (blue) and showing the correct location of the tetrode (right, arrowhead). Scale bar, 1 mm. fmi, forceps minor of the corpus callosum; CPu, Caudate-Putamen; PL, prelimbic cortex; IL, infralimbic cortex. **c** Representative raw LFP signal from the mPFC (top), used to extract the theta (4–12 Hz, middle) and gamma (30–90 Hz, bottom) frequency components following band-pass filtering. **d** Color-coded spectrogram from a mouse LFP recording, showing theta (4–12 Hz) and gamma (30–90 Hz) oscillation powers during fear extinction retrieval. **e, f, i, j** Normalized (Norm.) power spectra of mPFC recordings from the center (C) and

periphery (P) of the open-field (OF) arena (**e, f**), as well as during tone and inter-trial interval (ITI) phases of the fear extinction retrieval session (**i, j**) in control (dark blue) and γ2$^{f/f}$ (light blue) mice. Theta (θ) and gamma (γ) frequency bands are indicated. Power spectra are averaged of all animals, and the curves and the shaded areas indicate mean±s.e.m. **g, h, k, l** Dot plots showing the normalized (Norm.) theta (**g, k**) and gamma (**h, l**) oscillation powers for the center (C) and periphery (P) of the open-field arena (**g, h**), and for tones and inter-trial intervals (ITI) during the fear extinction retrieval session (**k, l**) in control (dark blue) and γ2$^{f/f}$ (light blue) mice. Control mice: $N = 5$ and γ2$^{f/f}$ mice: $N = 5$; $p = 0.8797$ in center and $p = 0.8966$ in periphery for (**g**); $p = 0.4152$ in center and $p = 0.4343$ in periphery for (**h**); $p = 0.2222$ during tones (two-tailed Mann–Whitney U test) and $p = 0.1222$ during ITIs for (**k**); $p = 0.03$ during tones and $p = 0.017$ during ITIs for (**l**); two-tailed Student's $t$ test. All data are represented as mean±s.e.m. *$p < 0.05$, ns: not significant. 4-OHT: 4-hydroxytamoxifen. Source data are provided as a Source data file.

compared to the ITI in both control and γ2$^{f/f}$ mice (Supplementary Fig. 10b, f). This increase was observed across the entire session as well as at the different stages within the session (Supplementary Fig. 10d, e, h, i), indicating a robust network activation and an ability in both groups to discriminate the tone stimuli. This strong theta–gamma coupling serves as a hallmark of effective cognitive processing, especially since no significant changes in coupling were observed between the center and periphery of the arena in the open field task (Supplementary Fig. 9g, h). However, despite the clear increase in coupling

during tone presentation compared to ITI, there were no significant differences between control and γ2$^{f/f}$ mice in the tone/ITI coupling ratio, either for low-gamma or high-gamma oscillations (Supplementary Fig. 10c, g). The lack of coupling defects shows that the coordination of broad network interactions remains intact, despite a decrease in low-gamma power.

To further dissect the temporal dynamics of gamma power alterations, we analyzed the relative power of low- and high-gamma oscillations at different stages of the retrieval session: early,

intermediate, and late (Fig. 7c, d). We found that low-gamma power was significantly reduced during the early and intermediate stages in γ2[f/f] mice compared to controls (Fig. 7c) during tone presentations and ITI, while it remained unaffected during the late stage. These early and intermediate reductions in low-gamma power correspond with the higher freezing levels observed in γ2[f/f] mice at these stages (Fig. 1f), suggesting a temporal relationship between low-gamma disruption and behavioral deficits. In contrast, high-gamma power exhibited a trend toward reduction that did not reach statistical significance and remained comparable between control and γ2[f/f] mice throughout all three stages of the retrieval session (Fig. 7d), further supporting the conclusion that the observed alterations are more pronounced in the low-gamma frequency range.

In summary, our findings suggest that low-gamma oscillations in the mPFC play a more critical role than high gamma oscillations in supporting fear extinction retrieval. The reduction in low-gamma power, particularly during the early and intermediate stages of the retrieval session in γ2[f/f] mice, points to its essential role in enabling neuronal network activity required for the successful fear extinction learning and memory. This reduction, alongside the lack of changes in theta–gamma coupling, is consistent with an intracortical mPFC network dysfunction, resulting from PV interneuron impairments which are key for intracortical projections.

## Discussion

This study reveals that the early postnatal period is a critical window for interneuron-oligodendroglia interactions, during which disruption of OPC GABAergic synaptic signaling in γ2[f/f] mice leads to persistent impairments in local neuronal network function and cognitive performance in adulthood. Specifically, deficits in learning and memory during tone fear extinction are associated with disrupted low-gamma oscillations in the IL region of the mPFC. These alterations arise from early dysfunctions in interneuron-OPC signaling, leading to impaired PV interneuron myelination and disrupted neuronal E/I balance. Our findings show the necessity of this developmental phase for oligodendroglia and PV interneuron myelination to sustain adult low-gamma oscillations that are critical for tone fear extinction learning and memory. Overall, this study emphasizes the importance of early postnatal PV interneuron myelination in the maturation of inhibitory circuits and its role in maintaining cortical low-gamma rhythms essential for specific cognitive functions. Consistent with this, adult-specific induction of γ2 deletion had no effect on extinction behavior, suggesting that this signaling pathway is only functionally relevant during early postnatal development. This is likely due to the fact that functional γ2-containing GABA-A receptor-mediated synaptic inputs are not present in adult OPCs[23,24].

Defects in OL lineage cells and myelination are increasingly associated with a variety of cognitive deficits[1]. For example, social isolation, whether occurring immediately after weaning or during adulthood, results in hypomyelination of the mPFC[38,39], while early maternal separation impairs emotional responses and object recognition in adult mice, linked to premature OPC differentiation in this cortical region[40]. Proper myelination is also critical for emotional regulation and memory processes, including fear conditioning and extinction, as disruptions in de novo OL production or myelin formation prolongs fear responses and hampers the consolidation of contextual fear memories[2,3]. However, while cognitive impairments are often attributed to pronounced hypomyelination or demyelination, the impact of subtle, cell-type-specific dysmyelination, such as that in PV interneurons, remains largely unexplored. PV interneurons play a key role in regulating gamma oscillations and maintaining the E/I balance, both of which are essential for coordinated neuronal network function and cognitive performance[10,12,27]. Our findings suggest that even mild myelination defects in PV interneurons during critical developmental windows can lead to persistent alterations in inhibitory

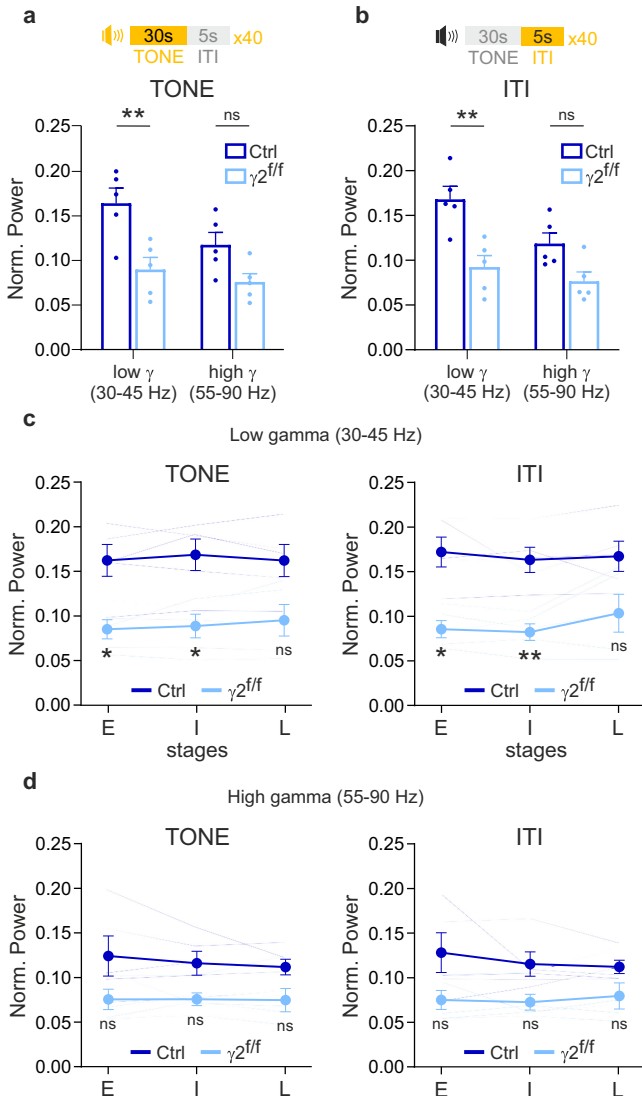

**Fig. 7 | Disrupted low-gamma oscillations during fear extinction retrieval in γ2[f/f] mice. a, b** Dot plots of the relative power of low-gamma and high-gamma oscillations during tone (**a**) and inter-trial interval (ITI) (**b**) in control (dark blue) and γ2[f/f] (light blue) mice. Control mice: N = 5 and γ2[f/f] mice: N = 5; p = 0.0033 (low-gamma) and p = 0.0978 (high-gamma) for (**a**); p = 0.0013 (low-gamma) and p = 0.0629 (high-gamma) for (**b**); two-way ANOVA test followed by post-hoc Sidak multiple comparisons test. **p < 0.01, ns: not significant **c, d** Comparisons of the normalized (Norm.) power of low-gamma (**c**) and high-gamma (**d**) oscillations during tone (left) or inter-trial interval (ITI; right) in control and γ2[f/f] mice at early (E), intermediate (I) and late stages (L) of the extinction retrieval session (average responses during 13 tones for each stage, see Supplementary Fig. 2i). Individual animal responses are shown by faint traces. Control mice: N = 5 and γ2[f/f] mice: N = 5; p = 0.026/0.0237/0.0860 at early, intermediate and late stages during tones and p = 0.0109/0.0061/0.1398 at early, intermediate and late stages during ITIs for low-gamma (**c**); p = 0.2781/0.1078/0.1470 at early, intermediate and late stages during tones and p = 0.2159/0.1005/0.2578 at early, intermediate and late stages during ITIs for high-gamma (**d**); two-way ANOVA test followed by a post-hoc Sidak multiple comparisons test. All data are represented as mean±s.e.m. *p < 0.05, **p < 0.01, ns: not significant. Source data are provided as a Source data file.

circuitry, impairing cognitive function. The alterations in low-gamma oscillations observed in our study, without significant changes in high-gamma oscillations, suggest that specific deficits in PV interneurons can selectively impact distinct gamma frequency bands. This cell-specific dysmyelination could represent a previously underestimated

mechanism underlying cognitive deficits and emotional dysregulations present in neurodevelopmental and psychiatric disorders, where disruptions in PV interneuron function and gamma oscillations are commonly implicated[1]. Our findings open the possibility that subtle myelination abnormalities in PV interneurons may contribute to the E/I imbalance and network dysfunctions characteristic of conditions such as schizophrenia and autism spectrum disorders[1]. The contribution of oligodendroglia to these conditions has likely been underestimated and deserves further investigation.

Recent studies in healthy mice demonstrate that behavioral training and stimuli can induce subtle changes in myelination. Motor-skill learning induces myelin remodelling, including the elongation of nodes of Ranvier[41], highlighting the importance of a fine-tuned myelination during learning processes. Furthermore, the length of nodes of Ranvier is modulated in distinct ways by interventions like repetitive transcranial magnetic stimulation or spatial learning[42]. Despite these insights into activity-dependent delicate myelination changes, the role of oligodendroglia and myelin in modulating neuronal network activity and cortical oscillations during the execution of cognitive processes in health and disease is poorly understood. A recent study using cuprizone-treated mice has shown that prolonged impairments in social and spatial working memory, as well as in cognitive flexibility, persist after remyelination. These impairments correlate with long-term desynchronized activity between the hippocampus and the mPFC, characterized by lasting disruptions in gamma synchronization and ripple/slow oscillation coupling, a condition that develops during the demyelination phase and is only partially restored during remyelination[43]. Furthermore, fiber photometry has revealed that animals unable to form new myelin exhibit a lack of the time-dependent modulation in mPFC neuronal activity seen in control animals, which initially show a decrease and then a progressive increase in activity during transitions from mobility to freezing[3]. Active oligodendrogenesis and myelination are therefore required for the proper evolution of prefrontal dynamics associated with fear memory recall. Moreover, the disruption of de novo production of OLs immediately before contextual fear conditioning also impairs the coupling between spindle oscillations in the prefrontal cortex and sharp wave ripple oscillations in the hippocampus, independent of myelin deficits[2]. Therefore, both OL production alterations and myelin deficiencies are linked to impaired neuronal responses in brain regions activated during cognitive tasks.

While the recognition of the involvement of oligodendroglia in cognition is growing, the mechanisms by which these cells influence specific neuronal subtypes and network dynamics during the execution of a cognitive task remain elusive. Our results demonstrate that early disruption of OPC GABAergic signaling results in a selective dysmyelination of PV interneurons, without major changes in overall oligodendrogenesis or myelination, and altered low-gamma oscillation activity during tone fear extinction. This defect does not uniformly impact neuronal networks or cognitive processes, as high-gamma oscillations show no significant change, and no impairments were observed in fear acquisition, context fear extinction, or exploratory behavior in the open field. The specific effects observed in our study may be explained by the fact that myelination characteristics such as patterns, myelin content, internode and node lengths vary across specific neuronal identities and for a specific cell type in different brain areas[21,44], which can lead to different consequences. This selective effect may arise from differences in myelination requirements in mPFC subregions, with the IL region depending on low-gamma oscillations critical for extinction memory, while the prelimbic (PL) region, involved in fear learning, may rely on different oscillatory patterns and thus be less affected by PV interneuron dysmyelination. In any case, our findings emphasize the critical role of precise myelination patterns of specific neuronal subtypes in maintaining proper network dynamics during distinct cognitive functions. While neuronal contributions to

cognitive processes are undisputable, our study positions oligodendroglia and myelination as pivotal elements driving the construction of prefrontal-dependent low-gamma oscillations necessary for a specific cognitive process. In addition to oligodendroglia, other types of glial cells, such as astrocytes, also play a significant role in regulating neuronal network dynamics. Recent findings suggest that GABAergic signaling to astrocytes, although non-synaptic in this case, also influences low-gamma oscillations without impacting high-gamma oscillations[45], further strengthening the importance of glial-neuronal interactions in maintaining cognitive functions. These findings highlight the importance of glial cells in fine-tuning network oscillations important for higher-order cognitive functions.

While slow oscillations like theta oscillations result from the coordinated activity of widely distributed neuronal ensembles, fast-frequency oscillations like gamma oscillations are considered a local phenomenon, involving smaller tissue volumes[1]. Cortical PV interneurons contribute to various frequency oscillations, but their role in gamma oscillations is crucial. This is due to their ability to discharge action potentials at high frequencies and their extensive local synaptic connectivity with surrounding excitatory neurons, which facilitates rapid and synchronized neuronal activity essential for gamma oscillations[11–13]. Our study revealed that PV interneuron dysmyelination in $\gamma2^{f/f}$ mice are associated with a reduced inhibitory drive causing an E/I imbalance and alterations of low-gamma oscillations in the IL region during tone fear extinction retrieval. A critical involvement of PV interneurons in low-gamma activity during cognitive processes is supported by a previous study showing a rescue of low-gamma oscillation alterations and social deficits by the optogenetic stimulation of PV interneurons at 40 Hz in a model of autism spectrum disorders characterized by a Neuregulin-3 mutation[46]. Moreover, the optogenetic activation of PV interneurons at 30 Hz allows for the preservation of gamma frequency oscillations in control, but not in mice with widespread demyelination[8]. In these models as well as in $\gamma2^{f/f}$ mice, a decreased PV interneuron firing rate is also observed[8,18,46]. Our findings uniquely highlight the role of PV interneuron myelination in sustaining low-gamma oscillations during specific cognitive tasks, aligning with previous studies that explore the function of these interneurons in cortical processing.

PV interneuron myelination remains particularly plastic throughout life, making these neurons more susceptible to myelin deficiencies. For instance, monocular deprivation results in experience-dependent adaptive myelin remodeling specifically in cortical PV interneurons, but not in glutamatergic projection neurons[29]. Furthermore, myelination defects occur specifically in PV interneurons of the mPFC in a rat model displaying schizophrenia-like behaviors[47], and demyelination induces selective vulnerability of GABAergic interneurons in both progressive multiple sclerosis and a mouse model of local cortical demyelination[48]. In a transgenic mouse, mild hypomyelination due to the loss of oligodendrocyte ErbB receptor signaling results in a decreased PV expression and a reduction in inhibitory but not excitatory synaptic connections, causing an E/I imbalance in L2/3 neurons of the auditory cortex[49]. Thus, PV interneuron myelination contributes to the pathological mechanisms underlying various neurological disorders, making it a novel and attractive pharmacological target[50]. Interestingly, it was previously shown that chemogenetic activation of hippocampal PV interneurons of adult mice subjected to a modified fear conditioning protocol with severe electric foot shocks reduced fear generalization and restored myelination of PV interneuron axons[51]. Nevertheless, altering early interneuron-oligodendroglia interactions, myelination, and neuronal plasticity during critical periods can generate lasting cognitive and behavioral effects that may be challenging to reverse. In our study, targeted interventions to increase PV interneuron activity or enhance global myelination in $\gamma2^{f/f}$ mice did not repair the defect in PV interneuron internode length or lead to improvements in fear extinction memory. This suggests that, once

early interneuron-OPC interactions are compromised, subsequent attempts to enhance myelination or promote PV interneuron activity are ineffective in reversing the resulting cognitive deficits. The critical developmental window for these interactions, occurring transiently in the first two postnatal weeks[16], appears crucial for establishing proper cognitive development and function. Preserving the integrity of transient interneuron-OPC interactions during this period appears essential to prevent long-lasting and irreversible cognitive deficits. It should be noted that most psychiatric treatments begin after the onset of symptoms, which often occur well after these critical periods, making it challenging to reverse such early-established myelination deficits. Additional evidence suggests that early but transient dysfunctional events of interneurons can be deleterious to cortical function in adulthood. Kessaris et al. (2021)[52] demonstrated that an aberrant overproduction of PV interneurons during embryogenesis has lasting effects that persist into critical periods of cortical development, despite the normalization of cell density postnatally. This transient excess of interneurons during early development results in behavioral abnormalities in adulthood, indicating that the timing of interneuron clearance is essential for the proper maturation of neural circuits that underlie adult behavior. In humans, early life adversity correlates with increased OPC-mediated perineuronal nets production around PV interneurons in the ventromedial PFC; as perineuronal nets regulate the closure of developmental windows of cortical plasticity, this potentially limits cortical circuit neuroplasticity and promotes maladaptive behaviors[53]. Together, these findings emphasize the challenges in reversing cognitive deficits once developmental processes are disrupted. An interesting line for future research in myelin biology would be to explore how abnormalities in PV interneurons and OL lineage synergize, potentially increasing the risk of neurodevelopmental and psychiatric disorders.

## Methods

### Transgenic mice and induction of Cre-mediated recombination

All experiments adhered the European Union and institutional guidelines for the care and use of laboratory animals. Ethical approval was obtained from the French ethical committee for animal care of university Paris Descartes (Committee N°CEEA34) and the Ministry of National Education and Research (Project No: 202010131827131). The experimental approach involved *NG2cre^{ERT2+/-};Gcamp3^{f/f};γ2^{f/f}* mice (γ2^{f/f} mice) to inactivate γ2-GABA_A receptor-mediated synapses in OPCs while the *NG2^{creERT2+/-};Gcamp3^{f/f}* mice were used as controls[18,22]. Genotyping was conducted by PCR using primers specific to the different alleles, and Cre expression was induced from P3 to P5 by daily intraperitoneal injections of 0.2 mg of 4-OHT (Tocris Bioscience) as previously described[22]. The Cre expression in adult mice (P110-P112) was induced two weeks prior to the fear conditioning test by administering tamoxifen (1 mg in Miglyol oil, Caesar & Loretz, Germany) intraperitoneally once daily for three consecutive days. Animals were housed in controlled conditions with 12 h light/dark cycle, ad libitum access to food and water, and in average ambient temperature of 21 °C and 45% humidity. Both females and males were used at different postnatal stages from P10 to P134.

### Behavioral assays

Anxiety and locomotion were assessed in a 50 cm × 50 cm × 45 cm open field arena, where mice were placed for 10 min under a 50 Lux illumination. An infrared camera monitored animal movements and the ViewPoint behavioral analysis software processed the data (version 5.31.0.8, ViewPoint, Behavior Technology, France). The fraction of time spent in an outer area of 2500 cm² over an inner square area of 900 cm² was calculated. The open field test was used to assess locomotion by measuring distance traveled, velocity, time spent in the inner (center) or outer (periphery) area, and number of crossings in the center area.

The fear conditioning paradigm consisted of four consecutive days of behavioral testing in a chamber of 30 cm × 25 cm × 25 cm made of two aluminum sidewalls, acrylic front and rear walls and a floor of metal bars of 3.2 mm diameter separated by 7.9 mm. An infrared camera, controlled by a computerized system interface, monitored behavior and freezing (percentage of time of immobility) using the Videofreeze software (version 2.7.2.117, MED Associates, St. Albans, VT, USA). On day-1 (fear conditioning), mice (aged P95-P103) were individually placed into a dark conditioning chamber, odorized with 1% carvone (mint odor). After a baseline period of 3 min, animals received five pairings of an auditory conditional stimulus (CS; 30 s, 5 kHz, 85 dB) with the unconditioned stimulus (US) applied at the end of the tone (2 s, 0.5 mA foot-shock) with a 2 min ITI period. On day-2 (contextual fear extinction), mice were introduced in the same conditioning chamber, odorized with carvone, and allowed to explore for 18 min without presentation of CS or US. On day-3 (tone fear extinction), mice were placed in chambers with a modified context and odor (0.5% isopentyl acetate; banana odor). After a baseline period of 3 min, the CS alone was presented 40 times (30 s), separated by 5 s ITIs. On day-4 (tone fear extinction retrieval), mice were subjected to the same protocol of day 3. The open-field arena and fear conditioning chambers box were wiped with 70% ethanol and air-dried between mice.

Freezing behavior was automatically quantified in VideoFreeze software (v2.27). Animal movement was analyzed with the default settings (motion threshold: 18 arbitrary units, sample rate: 30 frames per second, linear method, minimum freeze duration: 30 frames), with freezing defined as immobility lasting at least 1 s. Freezing responses were calculated across inter-trial intervals (ITIs) on day 1, averaged across periods of 2 min on day 2, and those on day 3 and day 4 were calculated using the average freezing response to five consecutive CS (5-trial blocks). The tone fear extinction retrieval session (day 4) was divided into 3 stages (early, intermediate and late), each composed of 13 CS presentation and 13 ITIs for analysis of the behavioral and in vivo electrophysiology data. In this case, the last CS and ITI was excluded from the analysis. Since no sex differences were observed between male and female mice in both control and γ2^{f/f} groups (Supplementary Fig. 3a–h), data from both sexes were combined for each group.

The startle response was assessed using a startle apparatus (SR-LAB Startle Response System, v1.0.1.4, San Diego Instruments). The mice (P120) were exposed to tone stimuli of varying intensities (measured in dB) lasting 20 ms, with an inter-trial interval (ITI) ranging from 10 to 20 s, and under a constant background noise of 64 dB. Following a 5-min acclimation period in this environment, the tone stimuli were presented in random order to the animals. The startle response was quantified by analyzing both the maximal and mean startle amplitudes in response to the randomly applied tone stimuli.

The apparatus used for the elevated plus maze comprise two open and two perpendicular closed arms (37 × 10 cm each one), elevated 50 cm above the floor (version 5.31.0.8, ViewPoint, Behavior Technology, France). The closed arms had a plexiglass wall with a height of 15 cm. The mice (P100-P125) were located at the center of the maze for 4 min for free exploration. We measured the exploration time in the open arms as well as the distance traveled.

For the novel object recognition task, mice (P100-P125) were placed in the same Open Field chamber containing two identical objects (Familiar objects), separated by 20 cm, and allowed to explore for 5 min. Following a 2-h retention interval, the mice returned to the chamber for 5 min-exploration period, during which one of the familiar objects (always the left one) was replaced with a novel object. Object exploration was quantified by measuring the time spent interacting with each object, and a preference index (PI) was calculated as follows: PI = Time with Novel Object/(Time with Novel Object + Time with Familiar Object).

## Surgical implantation of the microdrive

A custom microdrive was fashioned using a 10 ml syringe cut to a length of 1.5 cm. This allows the Neuralynx EIB-16 headstage to be glued to the inside, serving as a base for a mini-Faraday cage. The cage was made by sticking a sheet of copper foil adhesive tape (NOLL Electronic, Germany) around the syringe and grounding it with a silver-soldered wire (Ø 203.2 µm, ref. 782000, A-M Systems). The conical bottom of the microdrive is made by cutting the bottom of a 10 ml Falcon® tube. This conical shape increased the surface in contact with the skull and eased the application of dental cement. The center of the conical base is drilled to glue a plastic guide for introducing the tetrode composed of 4 tungsten wires (Ø 50.8/101.6 µm) insulated with Teflon (ref. 795500, A-M systems). The tail of the tetrode along with the silver ground and reference wires were gently inserted into channels holes on the EIB-16-QC (Neuralynx), and pinned by small golden pins (Ref. 31-0603, Neuralynx). The fully assembled microdrive weighed less than 3 g.

For chronic implantation of the microdrive, adult control and γ2$^{f/f}$ mice (P100-P129) were deeply anaesthetized with isoflurane (4% for induction and 1.5% for maintenance, Vetflurane, Virbac). Local analgesics, including a solution of lidocaïne (Laocaïne, 7 mg/kg, Intervet-MSD), buprenorphine (BUPRECARE, 0.1 mg/kg; 2 mL/kg; Axience) and an ophtalmic gel (Ocrygel, DOMES PHARMA FR) were administered prior to or during the surgery. After fixing the animal in the stereotaxic frame (Kopf Instruments), the skin above the skull was disinfected with 70% ethanol and betadine solution (Vétédine, Vetoquinol, France) before incision. A hole was stereotaxically drilled to reach the IL region of mPFC using coordinates −0.4 mm mediolateral, +1.70 mm anterior-posterior relative to bregma. One additional hole in the interparietal region was performed to insert a screw overlying the cerebellum where the reference wire and the soldered silver ground wire were connected together. Then, the screw was sealed with a light-cure, cyanoacrylate-based adhesive (Loctite 4310) to ensure stability. Finally, the microdrive was fixed to the skull using resin-modified glass ionomer cement (Ketac Cem Plus Automix 3 M) and acrylic dental cement (Pattern Resin LS, GC America Inc). The surface of the exposed brain was covered with saline (0.9% NaCl) for the duration of the experiment. The skin was sutured (Mersilene, EH7147H, Ethicon) and a saline solution at 37 °C was administered subcutaneously (100 µL to 200 µL) to rehydrate the animal. Finally, mice were recovered from anesthesia in an environment at 37 °C before they returned to their home cages.

At the end of experimental procedures, mice were deeply anesthetized using an overdose of a solution of Ketamine (180 mg/kg, Clorketam 1000, Vetoquinol, France) and Xylazine (10 mg/kg, Rompum, Elanco France), and electrolytic lesions were made to make the electrode implantation site more visible. A current of 12 µA for 30 seconds was applied under gas anesthesia (isoflurane) through the implanted tetrode via the EIB-16-QC interface and the NanoZ impedance measurement system (Neuralynx, USA). Mice were then perfused with phosphate buffer saline (PBS) followed by 0.15 M phosphate buffer (PB; pH ≈ 7.3), containing 4% paraformaldehyde (PFA). Brains were kept 2 h in PFA, rinse three times in PB and incubated in sucrose 30% at 4 °C until sinking for cryoprotection. Then, brains were embedded with a film of OCT by gently rolling them (Microm Microtech), frozen in dry ice and stored at −70 °C. Coronal slices were cut with a cryostat (20 µm; Leica CM3050 S Cryostat), stained with DAPI and mounted. Sections were visualized using an epifluorescence microscope (Zeiss Axioplan 2) to confirm the recording location. Animals with the tetrode not clearly positioned in the mPFC were excluded from the analysis.

## In vivo electrophysiological recordings

Recordings were performed 7–15 days after surgery by using Neuralynx 16 channels preamplifier and 7 m tether with its data acquisition

software Cheetah (v6.3.2, Neuralynx, USA). LFPs were recorded at 30 kHz sampling rate. Animals were habituated to experimenter handling and to recording sessions in their home cage for 15 min twice a day during 3 days; recording duration were 10 min in the open-field and 27 min on day-4 of the fear conditioning test.

The analysis of mPFC oscillations were carried out with custom-made programs written in Python 3.9.7 (Spyder 5.1.5). All scripts used in this study are freely available at Angulo's lab gitlab repository (https://gitlab.com/d5674/c_ibaceta/psd-volt). The mPFC LFP signals were band-pass filtered from 1 Hz to 120 Hz (fifth-order IR Butterworth filter), 50 Hz notch filtered (Notch width 0.1 Hz) to minimize AC 50 Hz noise. For LFP analysis, the data was downsampled from 30 kHz to 1 kHz. Then, we performed a power spectrum analysis (PSD) using the Welch method from SciPy Python and Neurodsp library[54]. The parameters used were: 2048 FFTs, Hann window and 1024 overlapping points. Following the PSD calculation, we normalized the data obtained during the open-field exploration by averaging the values across all frequencies (from 1 Hz to 120 Hz). For fear extinction retrieval, we normalized the data by averaging the values of frequencies ranging from 0 Hz to 120 Hz during the baseline period (3 min). We defined three frequency bands, theta (4–12 Hz), low gamma (30–45 Hz) and high gamma (55–90 Hz), and obtained the PSD for each band range. For 3D plots depicting PSD over time and frequency, we used Neurodsp library[54].

The PAC analysis was conducted using the Hilbert Transform, extracting the phase of Theta and the amplitude of both low-gamma and high-gamma, with a FIT filter to isolate the three frequencies (using SciPy Python library). For the analysis during tone fear extinction, we considered three stages: baseline, sound and ITI. We further subdivided the sound and ITI in three phases based on trial numbers, according to their trials: early (from 1 to 13), intermediate (from 14 to 26) and late (from 27 to 39). We calculated PSD and PAC for each of these phases.

## In vivo fractalkine (FKN) infusions

In vivo FKN infusions were performed by adapting the protocol described by Watson et al. (2021)[34]. On the day of the surgery, FKN was dissolved in 0.2% BSA-PBS (75 µg/ml; Recombinant Soluble Mouse CX3CL1/Fractalkine, R&D Systems) or BSA in PBS (0.2%, Jackson ImmunoResearch). γ2$^{f/f}$ mice (P42-P46) were deeply anesthetized with isoflurane (4% for induction and 1.5% for maintenance, Vetflurane, Virbac). The surgical procedure and mouse recovery were similar to that described above. For the local delivery of these compounds, a hole was stereotaxically drilled to implant a canula, containing either FKN or BSA, into the right lateral ventricle using coordinates −1.000 mediolateral, −0.300 anterior-posterior, −2.500 dorsoventral relative to bregma. Finally, the micro-osmotic pump (model 1004, Alzet) was attached via a catheter to the cannula and placed in a subcutaneous pocket between the shoulder blades. Infusions were administered at a flow rate of 0.11 µL per hour for 25–28 days. In total, each mouse received 200 ng of FKN per day, a dose that promoted myelination in adult murine CNS[34,35]. At the end of the experimental procedure, mice were deeply anesthetized using an overdose of a solution of Ketamine (180 mg/kg, Clorketam 1000, Vetoquinol, France) and Xylazine (10 mg/kg, Rompum, Elanco France) and then perfused for immunostaining analysis.

## Viral injections and CNO protocol

Viral-mediated specific labeling of PV interneurons in the mPFC were performed by adapting the protocol described by Stedehouder et al., 2018[30]. Adult γ2$^{f/f}$ mice (P95-P105) were used for viral injections, receiving either pAAV-S5E2-dTom-nlsdTom for controls (Addgene, #135630) or pAAV-S5E2-Gq-P2A-dTomato (Addgene, #135635) for DREADD stimulation[31]. The surgical procedure and mouse recovery were similar to that described above. Viral particles were injected into

the mPFC through a 10 μl gastight syringe (model 1701, Hamilton) using a motorized micropump (Legato 130, KD Scientific) and silica needle (Ø 75/150 μm, PHYMEP), using coordinates −0.4 mm mediolateral, +1.70 mm anterior-posterior, −2.9 mm dorsoventral relative to bregma. Four weeks after surgery, all mice received intraperitoneal injections of CNO (Clozapine N-oxide (CNO) dihydrochloride, 1 mg/kg; Hello Bio) twice daily (approximately at 9:00 h and 18:00 h) for 14 consecutive days. At the end of the experimental procedure, mice were deeply anesthetized using an overdose of a solution of Ketamine (180 mg/kg, Clorketam 1000, Vetoquinol, France) and Xylazine (10 mg/kg, Rompum, Elanco France) and then perfused for immunostaining analysis.

### Patch-clamp recordings on acute slices

Mice were euthanized by cervical dislocation. Patch-clamp experiments were performed in the IL region of the mPFC using coronal slices of 300-μm-thick at P9-P13 or P59-P78 for OPCs in mPFC (Supplementary Fig. 1), P92-P124 for layer 2/3 pyramidal neurons (Fig. 2a, b) in mPFC; P132-P146 for layer 5 dTom⁺ PV interneurons and layer 2/3 dTom⁻ pyramidal neurons in mPFC (Supplementary Fig. 7), P126-P147 for principal neurons in basolateral amygdala (Fig. 2c, d) and CA1 pyramidal neurons in the hippocampus (Fig. 2e, f). Recorded cells were visualized with an Orca Flash 4.0 LT CMOS digital camera (Hamamatsu) and Micro-Manager software (version 1.4.21[55]) under an Olympus BX51 microscope equipped with a 40X fluorescent water-immersion objective. Excitation light to detect fluorescent OPCs was provided by the CoolLed pE-300 ultra-system (Scientifica, UK) and 470 and 525 nm filters for excitation and emission wavelengths, respectively.

Patch-clamp recordings were performed at RT using an extracellular solution containing (in mM): 126 NaCl, 2.5 KCl, 1.25 $NaH_2PO_4$, 26 $NaHCO_3$, 20 glucose, 5 pyruvate, 2 $CaCl_2$ and 1 $MgCl_2$ (95% $O_2$, 5% $CO_2$). Glutamatergic and GABAergic postsynaptic currents of IL layer 2/3 pyramidal neurons, BLA principal neurons with triangular somata, CA1 pyramidal neurons in the hippocampus and fluorescent IL layer 5 OPCs were evoked at −70 mV and 0 mV, respectively by a monopolar electrode (glass pipette; 100 ms pulse, 10 V; Iso-Stim 01D, npi electronic GmbH, Tamm, Germany), with a CsMeS-based intracellular solution containing (in mM): 125 $CsCH_3SO_3H$, 5.4-aminopyridine, 10 tetraethylammonium chloride, 0.2 EGTA, 0.5 $CaCl_2$, 2 $MgCl_2$, 10 HEPES, 2 $Na_2$-ATP, 0.2 Na-GTP and 10 $Na_2$-phosphocreatine (pH ≈ 7.3). The rate of stimulation was 0.2 Hz for neurons and 0.1 Hz for OPCs. Recordings were made without series resistance ($R_s$) compensation. $R_s$ was monitored during recordings and cells showing a change of more than 30% were discarded. Potentials were corrected for a junction potential of −10 mV. Acquisition was obtained using Multiclamp 700B and pClamp version 10.7 software, filtered at 4 kHz and digitized at 20 kHz. Evoked postsynaptic currents were analyzed off-line using pClamp version 10.7 software (Molecular Devices).

### Immunostaining and quantification

Mice were deeply anesthetized using an overdose of a solution of Ketamine (180 mg/kg, Clorketam 1000, Vetoquinol, France) and Xylazine (10 mg/kg, Rompum, Elanco France). For PV and GAD-67 immunolabelling, mice (P115-P125) were then perfused with PBS followed by 0.15 M phosphate buffer (PB; pH ≈ 7.3), containing 4% paraformaldehyde (PFA) and their brains were frozen in isopentane at −70 °C and cut into coronal slices with a cryostat (20 μm; Leica CM3050 S Cryostat). The slices were then fixed in 4% paraformaldehyde (PFA) for 10 min with 0.1 M phosphate buffer (PB) at 4 °C, washed first with PBS and then with tris buffered saline (TBS). Next, they were placed in a blocking buffer with 50 mM TBS, 5% goat serum, 0.4% bovine serum albumin (BSA) and 0.1% Triton x100 for 2 h at RT, and incubated with primary antibodies in a 5% serum, 10% BSA, 10% Triton and 50 mM TBS solution for three nights in a cold chamber (4 °C). Primary antibodies were rabbit polyclonal anti-PV (1:500; ref. PV-27, Swant) and mouse monoclonal anti-GAD67 (1:200; ref. MAB5406,

Sigma-Aldrich). Slices were washed three times in 50 mM TBS and 0.1% Triton x100 before they were incubated with secondary antibodies with 10% BSA, 10% Triton and 50 mM TBS at 4 °C for two nights. We used goat anti-rabbit DyLight-405 (ref. 35551; Thermofisher Scientific) and goat anti-mouse Alexa Fluor-546 (ref. A-11030, Thermofisher Scientific) at 4 °C for two nights at 1:200. Slices were washed three times with the same previous washing buffer and mounted with Vectashield mounting medium (ref. H-1000, Vectorlabs).

For other immunostainings, mice (age: P115-P125) were perfused and brains were kept 2 h in PFA, rinsed three times in PB and incubated in sucrose 30% at 4 °C until sinking for cryoprotection. Then, they were embedded with a film of OCT (Microm Microtech) by rolling them gently, frozen in dry ice and stored at −70 °C. Coronal slices were cut with a cryostat (20 μm; Leica CM3050 S Cryostat), permeabilized with 0.3% triton X-100 and 4% Normal Goat Serum (NGS) overnight and incubated with primary antibodies. For internode length analysis, the slices were incubated with anti-PV and anti-SMI-312 for four nights before the incubation with the respective secondary antibodies during 2 hours at room temperature. Then, they were first incubated with anti-MBP for two nights and then with its secondary antibody for 2 h at room temperature. For quantification of OPC and OL densities, the slices were incubated with anti-Olig2 and anti-CC1 for four nights and then with its secondary antibody for 2 h at room temperature. To quantify recombination efficiency, we followed a similar protocol using an anti-Olig2 and anti-GFP antibody to amplify the GCaMP3 reporter signal (GCaMP3 served solely as a reporter in this analysis). Primary antibodies used for immunohistochemistry were rabbit polyclonal anti-PV (1:500; ref. PV-27, Swant), mouse anti-SMI-312 (1:200; ref. 837901, BioLegend), rat monoclonal anti-MBP (1:200; clone 26; ref. AB7349, Abcam), rabbit polyclonal anti-Olig2 (1:400; ref. AB9610, Merck Millipore), mouse monoclonal anti-APC (CC1; 1:100; clone CC-1; ref. OP80, Merck Millipore) and chicken polyclonal anti-GFP (for detection of GCaMP3 here used as a reporter; 1:1000; ref. ab13970, Abcam). Slices with primary antibodies were washed three times in PBS and incubated in secondary antibodies coupled, respectively, to goat anti-rabbit DyLight-405 (ref. 35551, Thermofisher Scientific), goat anti-mouse Alexa Fluor-546 (ref. A-11030, Thermofisher Scientific), goat anti-rat Alexa Fluor-633 (ref. A-21094, Thermofisher Scientific) and goat anti-mouse DyLight-633 (ref. GTX76787; Genetex) at RT at 1:200.

For marker⁺ cell analysis, confocal images were acquired using 63x oil objectives and, for internode length and MBP analysis, confocal image stacks of 15−20 μm depth were acquired using HC PL APO 93x/ 1,30 motCORR glycerol objectives with an SP8 Leica confocal microscope and the LAS X software (version 3.5.7.23225). This objective is equipped with motorized correction for refractive index mismatch which allows to image deeper in a sample, thereby maintaining optimal resolution and signal intensity throughout the z-stack. With this configuration, we obtained a lateral resolution of ~200 nm, sufficient for precise measurement of internode lengths along PV+ axons in 3D. Images were then analyzed blind using the Fiji/ImageJ2 framework (NIH, version 1.53i[56]). Cells were counted from layer 1 to 6 in the IL region of mPFC with ROI manager tool (3−4 slices per animal). Each image containing 20 z-sections of 1 μm each. Cells that were partially cut from the image were excluded in three of the six sides of the cube to prevent border effects. PV⁺, GAD-67⁺, Olig2⁺ and CC1⁺ cell densities were calculated as the number of cells per volume (mm³). For the analysis of cell distributions, the x coordinate of each cell was extracted from FIJI/ImageJ2 and its position was normalized with respect to the total length of the layers of the mPFC, which can vary from one slice to another. Then, we calculated the frequency of cells in the normalized mPFC length with a binning of 0.1. To quantify MBP fluorescence coverage and intensity, we applied a mask generated from a binary (black & white) image in FIJI. This mask was created to specifically isolate regions with MBP signal, allowing for targeted analysis of fluorescence exclusively in these areas. To assess PV signal

**Article** https://doi.org/10.1038/s41467-025-66309-3

in axonal compartments of deep layers of the IL region, we applied a mask generated from SMI-312$^+$ axon labeling to isolate axonal regions in image stacks acquired with a 93x objective. Then, the intensity of PV$^+$ pixels was quantified to estimate the PV signal in axonal compartments. Internode analyses were systematically carried out in cortical layers V–VI of the IL region, where PV interneurons are most abundant. For each animal, internodes were detected in 1–8 slices (mean: $2.95 \pm 0.21$ slices per animal; $7.47 \pm 0.74$ internodes per animal across 3–5 animals per group). Only internodes flanked by two nude axonal regions were considered for analysis. Because such internodes are relatively sparse and difficult to detect in thick brain sections, the sampled segments were naturally distributed across the tissue, ensuring that quantifications reflected multiple independent axon populations rather than repeated measures from the same axon. Internode length was measured from semi-automatic 3D reconstructions of SMI-312$^+$/PV$^+$/MBP$^+$ segments obtained using Simple Neurite Tracer plugin in Fiji/ImageJ2 framework[18,57]. All quantifications were performed by an experimenter independent of the immunostainings and image acquisition, and blinded to the experimental conditions, ensuring unbiased analysis. Together, this strategy guarantees that the reported phenotype reflects consistent alterations across multiple animals, slices, and axons, rather than a sampling artifact.

## Statistics
All data are expressed as mean ± SEM. In the dot plots, individual mice (N) are represented by closed dots while repeated measures (n) are shown as open dots. Statistical analysis were performed using Graph-Pad Prism (v10.2.2; GraphPad Software Inc., USA), and JASP (version 0.16.4; JASP Team, 2022, Amsterdam, The Netherlands). Each group of data was first subjected to Shapiro-Wilk normality test. According to the data structure and variance equality, two-group comparisons were performed using a two-tailed Student's $t$ test, a two-tailed Welch's t test or a two-tailed Mann–Whitney U test for unpaired samples. We also used two-tailed (default output of JASP) linear mixed models with likelihood ratio tests to account for repeated measures within each animal. Multiple group comparisons were performed using the parametric two-way ANOVA followed by a post-hoc Sidak, Tukey's or Fishers Least Significant Difference (LSD) multiple comparison tests, as well as mixed-effects model followed by a Sidak's multiple comparison test for situations where data were not normality distributed. Differences were considered significant when $P < 0.05$.

## Reporting summary
Further information on research design is available in the Nature Portfolio Reporting Summary linked to this article.

## Data availability
The data generated and analyzed during this study are included in the manuscript and supplementary data. Source data are provided with this paper. The raw datasets are available for research purposes from the corresponding author upon request. Source data are provided with this paper.

## Code availability
Psd-volt Python scripts are hosted at https://gitlab.com/d5674/c_ibaceta/psd-volt and published under a Free Software GNU GPLv3 license.

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

## Acknowledgements

We thank the NeurImag platform, Phenobrain and the animal facility of IPNP and their funding sources (Fédération pour la Recherche Médicale, Fondation Leducq). We would also thank Ludivine Therreau from Phenobrain as well as Beata Turanska and Julie Cognet in the team for their technical assistance, Steven A. Kushner and Karim Benchenane for insightful discussions on the project, Najate Benamer, Marie Vidal, Yana Kibalnyk, Philippe Faure and Elise Bousseyrol for help at the preliminary phase of the study. M.C.A. is a CNRS (Centre National de la Recherche Scientifique) investigator. A.V. and M.C.A. were supported by an Era-Net Neuron grant (CIHR and ANR under the frame of Neuron Cofund, No. 161466 and No. R19068KK for A.V. and M.C.A., respectively). A.E.S.W. was supported by CIHR CGSD. A.V. was supported by CIHR grants (PS 166120 and PJT 195940), MS Canada grant (1249541), the Scottish Rite Charitable Foundation of Canada grant as well as Canada Research Chair in Neural Stem Cell Biology and Sloan Research Fellowship in Neuroscience. M.C.A. was supported by grants from Fondation pour la Recherche Médicale (FRM, EQU202103012626 and MND202310017891), ANR (ANR CoLD: ANR-20-CE16–0001-01; ANR Myelex: ANR-21-CE37–0020-01), Fondation de France (00147199/WB-2023-50772) and Fédération pour la Recherche sur le Cerveau (1R24183PNOPES).

## Author contributions

F.P. conducted the majority of the experiments and performed data analysis of the study. C.I-G. developed the Python code for in vivo electrophysiological analysis and contributed to behavioral tests and data analysis. H.K. performed patch-clamp recordings of OPCs and

neurons in different brain regions as well as with virus injections. C.I-G. and H.K. equally contributed to this work as second co-authors. C.P. carried out behavioral experiments and immunostainings. A.E.S.W. and A.V. performed and assisted with FKN infusions. M.C.A. supervised the project, contributed to data analysis, and drafted the initial version of the manuscript. All authors edited and approved the manuscript.

## Competing interests

The authors declare no competing interests.
