## [Transparent Peer Review file · Nature Communications]

Prefrontal gamma oscillations and fear extinction learning require early postnatal interneuron-oligodendroglia communication

Corresponding Author: Dr Maria Cecilia Angulo

Version 0:

Reviewer comments:

Reviewer #1

(Remarks to the Author)

The study reports that early postnatal GABAergic signaling between interneurons and OPCs promotes myelination of parvalbumin interneurons. When this signaling was disrupted by suppressing GABAergic synapses in OPCs in transgenic mice, internodal length on PV+ neurons increased, tone fear extinction was impaired, and low-gamma power was reduced in mPFC. Increasing PV interneuron activity enhanced myelination but did not rescue the deficits, suggesting long-term consequences of impairing GABAergic-OPC signaling in early development.

The study concerns an important area of current research in the field of myelin plasticity. It expands understanding of activity-dependent myelination and OPC development in nervous system development and function, and in particular with reference to GABA signaling, myelination of interneurons, and the importance of oligodendroglia in neural oscillations and synchronization. The effects are robust, the data analysis is appropriate, appropriate controls are performed, and the results and conclusions are well supported. The evidence is comprehensive, including anatomical, electrophysiological, and behavioral data. The text and figures are clear.

(Remarks on code availability)

Reviewer #2

(Remarks to the Author)

In this manuscript the authors continue to explore the consequences of disrupting early postnatal communication between OPCs and GABA neurons on myelination and behaviour. In previous studies, the authors had demonstrated that this early communication is critical for the myelination of fast-spiking interneurons and their function in the juvenile somatosensory cortex. In the current manuscript, the authors explore the same genetic model but now examine the effects on the mPFC and cognitive performance in adult animals. In line with their previous work, they observe behavioural impairments, in particular, in tone fear extinction learning, and this is accompanied by an increase in internode length on PV INs, impaired inhibition and E/I imbalance in the mPFC. These deficits were not rescued by either chemogenetic activation of PV interneurons in adult mice or administration of fractaline in adults which enhanced myelination. Adult mice with developmental disruption of the OPC-IN interaction showed deficiencies in low-gamma frequency power during fear extinction sessions.

While the manuscript is interesting, the findings offer only incremental advancements given the previous work of the authors.

Two significant issues remain unanswered:

1. To what extent is the phenotype caused by the early loss of OPC-IN interaction versus the loss of this interaction in adult OPCs? While their previous work had demonstrated that OPC-GABA currents are reduced in later stages of development in the mouse barrel cortex, this has not been explored in the mPFC. Also, expression of $\gamma 2$ is not entirely abolished in NG2

cells with advancing development, leaving open the possibility that OPC-IN GABA currents in adult animals may still exist in the mPFC. In order to confidently distinguish between developmental versus adult effects, the authors would need to show a lack of $\gamma 2$ -mediated GABA currents in adult OPCs in the mouse PFC or induce the conditional loss of $\gamma 2$ in adult mice prior to fear conditioning experiments.

2. Given that the genetic manipulation is brain-wide, the study would be greatly enhanced by conditional deletion of $\gamma 2$ specifically in the mPFC using viral Cre injections, followed by behavioural studies. Only then can the behavioural phenotype be confidently ascribed to deficiencies in the mPFC.

Other issues:

3. In the final figure the authors state that there is no effect of loss of OPC GABA currents on high-gamma frequency power, however, the data show a clear reduction that does not reach significance. Are the authors convinced by this result? It should be commented on that there is a trend towards a reduction in high gamma.

4. Related to point 3 above, in several figures, especially when it comes to morphological/immunohistochemistry experiments, the data represent $n=3$ animals and the effects studied have a huge variation among animals (e.g. Fig 2, 3, 4, 5 etc). Have the authors performed power analysis to ensure that their experiments are sufficiently powered? All statistical analysis should be performed on biological replicates (different mice) and not technical replicates (repeat measures from single mice). It's not clear whether this is what has been done in this case.

5. The authors attempt to rescue the fear conditioning deficit observed in the cKO mice by increasing neuronal activity in PV interneurons through DREADD-induced activation and by increasing myelination with fractalkine. The rationale behind these interventions is not clearly explained. For the DREADD experiment, reduced activity of PV interneurons in the cKO mice is demonstrated in a later figure therefore this should be rearranged to form a stronger rationale. Additionally, the authors have stated that OPC GABA currents disappear post development, therefore based on this, increasing neuronal activity will likely have no effect on PV adaptive myelination. For the fractalkine experiment, the authors observe that their cKO mice have an increased myelin sheath length, why, therefore would the authors hypothesise that increasing myelination further could rescue this effect? A better explanation of the rationale behind these experiments would strengthen the manuscript.

6. The authors should reference the Kaller et al 2024 paper that found blocking adult oligodendrogenesis causes changes in EEG spectral power "Ablation of oligodendrogenesis in adult mice alters brain microstructure and activity independently of behavioral deficits"

7. Image 2d: the bottom panel shows a different staining in red for PV between the images – with the red on the left showing stronger expression at the ends of myelin sheaths. Is this caused by different brightness between the two images?

(Remarks on code availability)

Reviewer #3

(Remarks to the Author)

(Remarks on code availability)

Reviewer #4

(Remarks to the Author)

The study by Dr. Angulo and colleagues explores the impact of altering the interactions between PV interneurons and OPCs during early postnatal development on PFC myelination and cognitive functions in the adult. They report that KO of the GABAR $\gamma 2$ subunit from OPCs postnatally (induction at P3-P5) leads to several PFC phenotypes in the adults, including mild PV axon dysmyelination (primarily longer internodes) and E/I imbalance. Furthermore, the mice also show impairments in extinction of fear conditioning combined with altered gamma oscillations during extinction (but not during exploration). These deficits are described as irreversible, as interventions enhancing PV interneuron activity or promoting myelination failed to restore function. Based on these findings, the authors conclude that early interneuron-OPC synaptic signaling shapes mPFC circuits and associated fear extinction learning and memory.

The study addresses important questions with robust techniques, and the results are very intriguing. However, several important controls and clarifications are necessary to make the results and conclusions convincing.

Major points:

The authors claim their "study reveals that the early postnatal period is a critical window for interneuron oligodendroglia

interactions". However, since they did not check what happens if the KO is induced later, this study has not addressed the notion of a critical period. Since OPCs continue to be present in the adult brain and new myelinating oligodendrocytes are formed throughout life, this is an important question.

The efficiency of OPC recombination, a very critical piece of data, was not measured. It is not clear why the authors did not do that by quantifying the percent of NG2 cells expressing Gcamp3.

Since the Cre line should induce recombination in OPCs all over the brain, it is most likely altering OPC-interneuron communication in other regions. Given that the PFC integrates inputs from other brain areas, including areas related to cognitive function (e.g., hippocampus and amygdala), it seems likely that disruptions in circuits in other brain regions contribute to the functional phenotypes. Functional analysis of different brain regions is necessary to define the importance of the PFC circuits versus others.

The core of the study is the correlation between longer internodes and the behavioral/physiological phenotypes. However, the internodes were analyzed in only three mice per group, and the number of axons/internodes quantified per animal are not defined. This raises profound concerns about the robustness and generalization of the author's conclusion. Furthermore, internode lengths can vary with axonal subtype or cortical layer, and it is unclear in which PFC regions (or in which depth within slices) quantifications were made. For example, do they know if the phenotype reflects changes of all internodes in a few axons or a few internodes in many axons? Additionally, multiple measurements from the same mouse are not true biological replicates, as they are influenced by the same genetic and environmental conditions. More robust approaches, e.g. electron microscopy analysis of internodes and myelin phenotype, would help make the analysis stronger.

To interpret the negative results obtained with chemogenetics and fractalkine treatments it is essential to do the same experiment on control mice (NG2creERT-GCaMP3 floxed).

CNO treatment is assumed to activate PV interneurons via the hm3Dq DREADD system. However, it is unclear whether the intervention was successful without providing direct evidence. The study does not directly demonstrate that CNO treatment increased PV neuron activity. This weakens the interpretation of the results. Also, previous studies have reported that CNO can have off-target effects or be metabolized to clozapine, which could influence neuronal activity independently of DREADD activation. Verifying PV-specific activation would rule out these confounding factors.

The authors attempted to rule out auditory deficits as a confounding factor for fear conditioning by measuring the startle responses. This test uses suprathreshold sound levels and thus does not assess auditory function, it just measures the startle response (of course, if the mice are deaf, they could detect a problem). There are standard methods to measure auditory function in mice.

It is unclear why they used the GCaMP3 floxed allele mice without performing calcium imaging experiments. Calcium imaging could have provided critical, real-time insights into the functional dynamics of OPCs and their interactions with PV interneurons.

Other points:

- 1 – Methods: Description of the experiments performed to measure startle response is lacking.
- 2 – The representative images in the Extended Data Figure 4 suggest a reduction in PV axon density. This should be tested.
- 3 – On page 10, lines 183/184, the text incorrectly states that CNO treatment was applied in controls. It was only administered to $\gamma 2f/f$ mice.

(Remarks on code availability)

Version 1:

Reviewer comments:

Reviewer #2

(Remarks to the Author)

The authors have addressed all the concerns raised in my previous review. The current manuscript is very much improved and constitutes an elegant study into the role of PV interneuron myelination in finetuning in vivo inhibitory network dynamics underlying cognition.

(Remarks on code availability)

Reviewer #3

(Remarks to the Author)

(Remarks on code availability)

Reviewer #4

(Remarks to the Author)

The authors did an impressive amount of new experiments in a relatively short time and answered almost all of the reviewers' concerns.

(Remarks on code availability)

Reviewer #1 (Remarks to the Author):

The study reports that early postnatal GABAergic signaling between interneurons and OPCs promotes myelination of parvalbumin interneurons. When this signaling was disrupted by suppressing GABAergic synapses in OPCs in transgenic mice, internodal length on PV+ neurons increased, tone fear extinction was impaired, and low-gamma power was reduced in mPFC. Increasing PV interneuron activity enhanced myelination but did not rescue the deficits, suggesting long-term consequences of impairing GABAergic-OPC signaling in early development.

The study concerns an important area of current research in the field of myelin plasticity. It expands understanding of activity-dependent myelination and OPC development in nervous system development and function, and in particular with reference to GABA signaling, myelination of interneurons, and the importance of oligodendroglia in neural oscillations and synchronization. The effects are robust, the data analysis is appropriate, appropriate controls are performed, and the results and conclusions are well supported. The evidence is comprehensive, including anatomical, electrophysiological, and behavioral data. The text and figures are clear.

We thank the reviewer for the positive feedback and the constructive comments. We are delighted that this reviewer considers our work as a meaningful contribution to the field. The recognition of the robustness of our results and the clarity of our presentation is greatly appreciated.

Reviewer #2 (Remarks to the Author):

In this manuscript the authors continue to explore the consequences of disrupting early postnatal communication between OPCs and GABA neurons on myelination and behaviour. In previous studies, the authors had demonstrated that this early communication is critical for the myelination of fast-spiking interneurons and their function in the juvenile somatosensory cortex. In the current manuscript, the authors explore the same genetic model but now examine the effects on the mPFC and cognitive performance in adult animals. In line with their previous work, they observe behavioural impairments, in particular, in tone fear extinction learning, and this is accompanied by an increase in internode length on PV INs, impaired inhibition and E/I imbalance in the mPFC. These deficits were not rescued by either chemogenetic activation of PV interneurons in adult mice or administration of fractaline in adults which enhanced myelination. Adult mice with developmental disruption of the OPC-IN interaction showed deficiencies in low-gamma frequency power during fear extinction sessions.

While the manuscript is interesting, the findings offer only incremental advancements given the previous work of the authors.

We thank the reviewer for his thoughtful comment. While our study builds on our previous work, it advances the field by uncovering a previously unrecognized role of PV interneuron myelination in fine-tuning in vivo inhibitory network dynamics underlying cognition. Importantly, we show that early-established myelin defects are difficult to reverse, illustrating their long-lasting impact. Given that the role of PV interneuron myelination remains poorly understood, and considering the strong links between PV interneuron dysfunction, gamma oscillation deficits, and neuropsychiatric conditions such as schizophrenia and autism, we believe our findings provide a novel framework for investigating oligodendroglial contributions to these disorders. Furthermore, by linking myelin deficits in specific axons to network oscillatory activity during the performance of a specific behavioral task, our study offers an essential step toward clarifying the functions of myelin in the brain.

Two significant issues remain unanswered:

1. To what extent is the phenotype caused by the early loss of OPC-IN interaction versus the loss of this interaction in adult OPCs? While their previous work had demonstrated that OPC-GABA currents are reduced in later stages of development in the mouse barrel cortex, this has not been explored in the mPFC. Also, expression of $\gamma 2$ is not entirely abolished in NG2 cells with advancing development, leaving open the possibility that OPC-IN GABA currents in adult animals may still exist in the mPFC. In order to confidently distinguish between developmental versus adult effects, the authors would need to show a lack of $\gamma 2$ -mediated GABA currents in adult OPCs in the mouse PFC or induce the conditional loss of $\gamma 2$ in adult mice prior to fear conditioning experiments.

To address this important point, we performed both of the suggested experiments:

First, we assessed GABAergic currents in adult OPCs from mice in which $\gamma 2$ subunits was deleted during early postnatal development. We recorded I_{GABA} in layer 5 OPCs in acute mPFC slices from 2-months control and $\gamma 2$ mice. As expected from previous reports (Balía et al., 2015, Cereb Cortex; Passlick et al., 2013, J Neurosci), adult OPCs exhibit evoked GABAergic currents. These currents were insensitive to zolpidem, the positive allosteric modulator selective for $\gamma 2$ -containing GABA-A receptors, and were detectable at similar levels in the adult in both groups. This pharmacological profile as well as the presence of similar GABA currents in control and $\gamma 2$ mice indicates the absence of functional $\gamma 2$ subunits in adult OPCs. These new data are now included in the revised manuscript (new Supplementary Fig. 1a-d, page 6, Line 104-112) and confirm that $\gamma 2$ -mediated GABAergic synaptic input is developmentally restricted and absent in adult OPCs in the mPFC.

Second, we conducted fear conditioning experiments in $\gamma 2$ mice that were either not injected with tamoxifen (thus retaining normal $\gamma 2$ expression) or were injected in adulthood to induce conditional Cre expression specifically in adult OPCs. These mice showed no behavioral differences compared to controls (new Supplementary Fig. 5a–e, black traces; page 8, lines 139-145), confirming that the genetic background, Cre transgene, and floxed alleles do not affect extinction behavior. Moreover, in contrast to the significant impairments in tone extinction observed following early postnatal $\gamma 2$ subunit deletion, adult-specific induction of Cre recombination did not result in any behavioral difference compared to non-injected $\gamma 2$ mice (new Supplementary Fig. 5f-b; page 8, lines 145-150, page 20, lines 426-430; page 26, lines 566-569). This indicates that recombination in adulthood has no physiological consequence on behavior. These findings confirm that the timing of recombination is critical, and that only early postnatal disruption of $\gamma 2$ -mediated interneuron–OPC signaling leads to the behavioral phenotype.

Taken together, these complementary experiments reinforce the conclusion that the behavioral phenotype arises from the disruption of interneuron-OPC communication during a critical developmental window, rather than from any changes in adulthood. This developmental specificity highlights the transient but critical role of early GABAergic signaling to OPCs in shaping long-term prefrontal circuit function and behavior.

2. Given that the genetic manipulation is brain-wide, the study would be greatly enhanced by conditional deletion of $\gamma 2$ specifically in the mPFC using viral Cre injections, followed by behavioural studies. Only then can the behavioural phenotype be confidently ascribed to deficiencies in the mPFC.

We appreciate the reviewer's suggestion to use conditional deletion of $\gamma 2$ specifically in the mPFC *via* viral Cre injections. While this approach would provide valuable insights, different teams have attempted to target OPCs using viral vectors, but achieving efficient infection in these cells remains highly challenging. For example, Chen et al. (2018, Cell Rep, PMID: 30355492) used retroviral infection of OPCs in the corpus callosum, but the efficiency was low and insufficient to perform behavioral studies.

This reflects the broader difficulty in targeting OPCs with viral approaches, which is why most studies on OPCs rely on transgenic mouse models rather than viral-mediated gene manipulation.

We agree, however, that further characterization would strengthen our conclusions, and we have used alternative approaches to address the reviewer's concern. As suggested by Reviewer #4, we expanded our functional analyses by using patch-clamp recordings in acute slices of the adult amygdala and hippocampus, the two other key regions involved in fear expression and extinction. In contrast to the mPFC, we did not observe significant changes in eEPSCs, eIPSCs or the E/I ratio in these other regions between control and $\gamma 2$ mice (see new Figure 2c-f, page 10, line 186-192, page 33, line 735).

To complement these findings, we performed more region-specific behavioral tests targeting these two regions. For the amygdala, we used the elevated plus maze to evaluate anxiety-related behaviors while for the hippocampus, we performed an object recognition test to assess recognition memory. No significant differences were observed between control and $\gamma 2$ mice in these tests (see new Supplementary Fig. 5f-g, page 10, line 193-205, and page 28, line 513-524). This aligns with our findings from the fear conditioning test, as the lack of effect on fear acquisition and contextual fear extinction already indicated no significant functional alterations in the amygdala and hippocampus, respectively.

These two new functional analyses further support a predominant mPFC involvement in the observed phenotype and provide a more comprehensive analysis of regional contributions and significantly strengthen the conclusions of our study.

Other issues:

3. In the final figure the authors state that there is no effect of loss of OPC GABA currents on high-gamma frequency power, however, the data show a clear reduction that does not reach significance. Are the authors convinced by this result? It should be commented on that there is a trend towards a reduction in high gamma.

We agree with the reviewer that the data suggest a trend toward reduced high-gamma power in $\gamma 2$ mice. However, the significant reduction in total gamma power in $\gamma 2$ mice (Fig. 6l) is primarily driven by low-gamma oscillations, which also showed a significant difference. In contrast, high-gamma power did not reach statistical significance during the entire session or at any stage during the session (Fig. 7a, b, d). In response to the reviewer's suggestion, we have revised the manuscript to explicitly emphasize the trend in high-gamma power in the abstract (page 2, line 34), Results section (page 19, lines 404-407), and Discussion section (page 22, line 480). These revisions provide a more nuanced interpretation of the data while maintaining the emphasis on the key findings related to low-gamma oscillations.

4. Related to point 3 above, in several figures, especially when it comes to morphological/immunohistochemistry experiments, the data represent $n=3$ animals and the effects studied have a huge variation among animals (e.g. Fig 2, 3, 4, 5 etc). Have the authors performed power analysis to ensure that their experiments are sufficiently powered? All statistical analysis should be performed on biological replicates (different mice) and not technical replicates (repeat measures from single mice). It's not clear whether this is what has been done in this case.

We appreciate the reviewer's concern regarding sample size and biological variability in our morphological and immunohistochemistry experiments. In our study, we performed multiple technical replicates on 3-5 biological (animal) samples per group for immunostaining experiments. While we acknowledge that a larger number of biological replicates would further enhance statistical power, our approach aligns with established practices in the field, where 3-5 slices per animal are commonly used, and many studies rely on imaging multiple fields from 3-5 mice.

Despite the inherent biological variability across animals, we observed remarkable consistency in the mean internode length of $\gamma 2$ mice across conditions: 40.2 μm in naive $\gamma 2$ mice, 38.67 μm after PBS infusions, and 40.15 μm following viral injections, compared to 31 μm in control mice. This consistency reinforces the robustness of our findings.

To ensure the reliability of our data, all morphological measurements were conducted in a fully blinded manner by an experimenter not involved in the immunostaining or confocal imaging (see Methods section, Page 36, lines 813-816). This approach minimizes potential bias and strengthens the validity of our findings despite biological variability. Moreover, our results are supported by our previous work (Benamer et al., 2020, Nat Commun PMID: 33051462), which involved a more extensive measurements of internode length in the somatosensory cortex, providing strong evidence for the biological phenomenon under investigation.

For statistical analysis, we employed linear mixed models, which are particularly well-suited for datasets with repeated measures. This approach allowed us to rigorously account for both within-animal (technical replicates) and between-animal (biological replicates) variability, ensuring a robust evaluation of our results. While we recognize the value of additional biological replicates, it is worth noting that the use of such statistical analysis is still relatively uncommon in the field for studies with similar sample sizes. We believe that the consistency of our findings, their coherence with our previous work, and the application of a robust statistical framework collectively support the validity of our conclusions.

5. The authors attempt to rescue the fear conditioning deficit observed in the cKO mice by increasing neuronal activity in PV interneurons through DREADD-induced activation and by increasing myelination with fractalkine. The rationale behind these interventions is not clearly explained. For the DREADD experiment, reduced activity of PV interneurons in the cKO mice is demonstrated in a later figure therefore this should be rearranged to form a stronger rationale. Additionally, the authors have stated that OPC GABA currents disappear post development, therefore based on this, increasing neuronal activity will likely have no effect on PV adaptive myelination. For the fractalkine experiment, the authors observe that their cKO mice have an increased myelin sheath length, why, therefore would the authors hypothesise that increasing myelination further could rescue this effect? A better explanation of the rationale behind these experiments would strengthen the manuscript.

We thank the reviewer for this insightful comment and appreciate the opportunity to clarify the rationale behind our intervention strategies. As suggested, we have rearranged the manuscript to present the electrophysiological evidence of reduced inhibition before introducing the DREADD intervention experiments (now new Figure 2). Regarding activity-dependent myelination, while OPC GABAergic inputs decline postnatally, recent studies demonstrate that activity-dependent myelination of PV interneurons exists in adulthood (Stedehouder et al., 2018, J Neurosci). Thus, DREADD activation may simultaneously address both the circuit-level deficit (by improving inhibition deficits, page 12, line 251-252) and myelin dysfunction (via activity-dependent repair mechanisms). Regarding the fractalkine experiment, our goal with this treatment was to test whether supporting oligodendrocyte maturation and myelination during a late, but still active myelination phase could compensate the structural and behavioral effects of early OPC synapse disruption. This has now been clarified in the revised manuscript (page 13; line 258-264).

6. The authors should reference the Kaller et al 2024 paper that found blocking adult oligodendrogenesis causes changes in EEG spectral power “Ablation of oligodendrogenesis in adult mice alters brain microstructure and activity independently of behavioral deficits”

We now cited this paper by adding in the introduction (page 3, line 57-59):

“...and ablation of oligodendrogenesis affects in vivo EEG brain activity even in the absence of task experience⁹.”

7. Image 2d: the bottom panel shows a different staining in red for PV between the images – with the red on the left showing stronger expression at the ends of myelin sheaths. Is this caused by different brightness between the two images?

We sincerely apologize for the issue with the staining presentation in previous Figure 2d, now Figure 3d. The difference in red staining intensity for PV between the images was not due to altered brightness but rather resulted from an incorrect PDF export option when converting the figure from CorelDraw to PDF. This unintentionally affected the image quality. We have now increased the resolution of all images and corrected the figure by using the appropriate export settings to ensure accurate and consistent representation of the data.

Reviewer #3 (Remarks to the Author):

We thank the reviewer for the feedback and appreciate the opportunity to collaborate with an Early Career Researcher through the Nature Communications initiative. We have carefully considered all comments of the reviewers and incorporated them into our revisions to improve the manuscript.

Reviewer #4 (Remarks to the Author):

The study by Dr. Angulo and colleagues explores the impact of altering the interactions between PV interneurons and OPCs during early postnatal development on PFC myelination and cognitive functions in the adult. They report that KO of the GABAR $\gamma 2$ subunit from OPCs postnatally (induction at P3-P5) leads to several PFC phenotypes in the adults, including mild PV axon dysmyelination (primarily longer internodes) and E/I imbalance. Furthermore, the mice also show impairments in extinction of fear conditioning combined with altered gamma oscillations during extinction (but not during exploration). These deficits are described as irreversible, as interventions enhancing PV interneuron activity or promoting myelination failed to restore function. Based on these findings, the authors conclude that early interneuron-OPC synaptic signaling shapes mPFC circuits and associated fear extinction learning and memory.

The study addresses important questions with robust techniques, and the results are very intriguing. However, several important controls and clarifications are necessary to make the results and conclusions convincing.

We thank the reviewer for the positive feedback and valuable suggestions, which we carefully considered to enhance the quality of our manuscript. Please, see below.

Major points:

The authors claim their study reveals that the early postnatal period is a critical window for interneuron oligodendroglia interactions. However, since they did not check what happens if the KO is induced later, this study has not addressed the notion of a critical period. Since OPCs continue to be present in the adult brain and new myelinating oligodendrocytes are formed throughout life, this is an important question.

We fully agree with the reviewer on this important point. We therefore performed additional experiments in which Cre expression was induced specifically in adult mice and then tested the fear conditioning paradigm. As detailed in our response to Reviewer #2, adult-specific induction of Cre expression did not result in significant differences in tone extinction behavior compared to non-injected $\gamma 2$ mice, which retain normal developmental $\gamma 2$ expression while carrying the same Cre and $\gamma 2$ alleles (new Supplementary Fig. 5b-e; page 8, lines 139-150, page 20, lines 425-429; page 26, lines 566-569). This contrasts with the significant deficits observed following $\gamma 2$ deletion during early postnatal development. These findings support the conclusion that interneuron-OPC interactions *via* $\gamma 2$ -containing GABA-A receptors are critical during a defined early postnatal window, but are not required in adulthood for the behavioral outcomes assessed.

Together with our new electrophysiological results showing a lack of functional $\gamma 2$ -mediated GABAergic currents in adult OPCs of the mPFC, (see also response to Reviewer #2; manuscript (new Supplementary Fig. 1a, 1d-e, page 6, Line 104-112), these data provide strong evidence that the early postnatal period represents a critical window for interneuron-oligodendroglial interactions relevant to long-term circuit function and behavior.

The efficiency of OPC recombination, a very critical piece of data, was not measured. It is not clear why the authors did not do that by quantifying the percent of NG2 cells expressing Gcamp3.

We thank the reviewer for this important point. While we had not initially quantified OPC recombination in the mPFC, we previously demonstrated efficient recombination in the somatosensory cortex (Balía et al., 2017, *Glia*; PMID: 28795438). In response to the reviewer's suggestion, we performed new immunostainings using Olig2 to label the OL lineage and GFP to enhance GCaMP3 signal detection. This allowed us to quantify recombination efficiency in the adult mPFC, which we found to be 65.5% in control mice and 63.4% in $\gamma 2$ mice. These new data have been added to the revised manuscript (new Supplementary Fig. 2a; Page 6, lines 112-116; Page 35, lines 771-773; 777-779).

Since the Cre line should induce recombination in OPCs all over the brain, it is most likely altering OPC-interneuron communication in other regions. Given that the PFC integrates inputs from other brain areas, including areas related to cognitive function (e.g., hippocampus and amygdala), it seems likely that disruptions in circuits in other brain regions contribute to the functional phenotypes. Functional analysis of different brain regions is necessary to define the importance of the PFC circuits versus others.

We acknowledge the reviewer's concern regarding the potential impact of $\gamma 2$ mice beyond the mPFC and performed additional functional analysis as suggested. As detailed in our response to Reviewer #2, patch-clamp recordings revealed no significant changes in eEPSCs, eIPSCs, or the E/I ratio in amygdala and the hippocampus (new Fig. 2c-f; page 10, line 186-192). Moreover, we performed more region-specific behavioral tests (elevated plus maze for the amygdala and novel object recognition for the hippocampus) and observed no significant differences between controls and $\gamma 2$ mice (new Supplementary Fig 5f-g; page 10, line 193-200). It can also be considered that the lack of effect on fear acquisition and contextual fear extinction (Fig. 1b, c) in our original fear conditioning paradigm already suggested that hippocampal and amygdala function remained largely intact. Overall, these findings confirm that the fear extinction deficits observed in our study mainly rely from disruptions in mPFC circuits rather than the amygdala and the hippocampus.

The core of the study is the correlation between longer internodes and the behavioral/physiological phenotypes. However, the internodes were analyzed in only three mice per group, and the number of

axons/internodes quantified per animal are not defined. This raises profound concerns about the robustness and generalization of the author's conclusion. Furthermore, internode lengths can vary with axonal subtype or cortical layer, and it is unclear in which PFC regions (or in which depth within slices) quantifications were made. For example, do they know if the phenotype reflects changes of all internodes in a few axons or a few internodes in many axons? Additionally, multiple measurements from the same mouse are not true biological replicates, as they are influenced by the same genetic and environmental conditions. More robust approaches, e.g. electron microscopy analysis of internodes and myelin phenotype, would help make the analysis stronger.

We thank the reviewer for highlighting the importance of a clear description of our internode analysis. We agree that the methodology was presented too concisely in the original submission and we now provide additional details to clarify the robustness of our approach (Page 36, line 804-816). Briefly, internode analyses were systematically performed in cortical layers V–VI of the IL region (15-20 μm depth), where PV interneurons are most abundant. Internodes were manually identified from distinct axonal segments located far apart within the slices. For each animal, internodes were blindly detected by a third experimenter in 1–8 slices (mean 2.95 ± 0.21 slices per animal; 7.47 ± 0.74 internodes per animal across 3–5 animals per group). Because identifiable internodes are sparse, this approach ensures that measurements reflect multiple independent axon populations rather than repeated measures from the same axon.

We would also like to emphasize that a comprehensive structural characterization of myelination deficits in PV interneurons has already been reported in the somatosensory cortex of $\gamma 2$ mice in our previous study (Benamer et al. 2020, Nat Commun, PMID: 33051462). In the present manuscript, our aim was not to reproduce this in-depth analysis, but rather to determine whether comparable alterations are also present in the adult mPFC. Repeating the same characterization would have introduced redundancy without substantially advancing our understanding. Moreover, while we agree that electron microscopy (EM) provides unparalleled ultrastructural resolution, it is not well suited for assessing internode lengths in a cell type-specific manner. Accurate length quantification would require serial sectioning or 3D reconstruction of PV+ axons, which is technically challenging, extremely resource- and time-intensive, and not easily scalable to the sample sizes required. In contrast, our approach allows direct identification of PV+ axons and reliable quantification of internode lengths in 3D across independent axons in a reproducible way. Importantly, our imaging was performed with a Leica 93x/1.30 GLYC motCORR objective, which compensates for sample-induced aberrations and provides optimal conditions for 3D imaging. With this setup, we achieve an effective resolution of ~ 200 nm, which is well within the range needed to reliably measure internode lengths. Thus, our methodology ensures robust detection of the relevant structural parameters (see Page 35, line 786-789).

To interpret the negative results obtained with chemogenetics and fractalkine treatments it is essential to do the same experiment on control mice (NG2creERT-GCaMP3 floxed).

CNO treatment is assumed to activate PV interneurons via the hM3Dq DREADD system. However, it is unclear whether the intervention was successful without providing direct evidence. The study does not directly demonstrate that CNO treatment increased PV neuron activity. This weakens the interpretation of the results. Also, previous studies have reported that CNO can have off-target effects or be metabolized to clozapine, which could influence neuronal activity independently of DREADD activation. Verifying PV-specific activation would rule out these confounding factors.

We appreciate the suggestion to examine NG2creERT-GCaMP3 control mice, but would like to clarify why chemogenetic and fractalkine experiments in these animals would not provide comparable results to our $\gamma 2$ mouse model. Regarding chemogenetic experiments, the functional impact of PV interneuron activation through hM3Dq receptors would differ fundamentally between the two models.

In $\gamma 2$ mice, where baseline inhibition is severely impaired, PV activation aims to compensate an already dysregulated system. In contrast, in control mice with normal inhibitory transmission, the same manipulation would simply enhance an already intact inhibitory transmission. These distinct starting points make the biological outcomes incomparable, as they represent different physiological contexts. Similarly, for fractalkine-mediated modulation of myelination, the baseline conditions differ dramatically between the models. In $\gamma 2$ mice, where altered myelination patterns and impaired inhibition exists, enhancing myelination may have completely different functional consequences compared to doing so in a normal system with intact myelin and circuits. Our experimental approach therefore focuses on validating these manipulations within the $\gamma 2$ -deficient system itself, which provides the most relevant framework for understanding their effects in the perturbed context we are studying.

We thank the reviewer for raising the important point concerning the specificity and efficacy of chemogenetic activation. To directly assess whether CNO treatment increased the activity of PV interneurons, we now performed whole-cell patch-clamp recordings in acute mPFC slices from $\gamma 2^{f/f}$ mice injected with the pAAV-S5E2-hM3D(Gq)-P2A-dTomato viral construct. These recordings allowed us to compare the electrophysiological responses of hM3Dq/dTomato-expressing PV interneurons with neighboring pyramidal neurons (lacking hM3Dq/dTomato) following CNO application. As shown in the new Supplementary Fig. 7 and described on page 13, lines 258-264, CNO application significantly enhanced the excitability of hM3Dq/dTomato-expressing PV interneurons, evidenced by a shorter latency to the first action potential and a higher mean firing rate in response to a depolarizing current ramp. In contrast, neighboring pyramidal neurons showed no such changes in excitability, confirming the selectivity of CNO action. These data provide direct functional evidence for successful chemogenetic activation of PV interneurons in our system.

Regarding concerns about potential off-target effects of CNO or its conversion to clozapine, our originally experimental design already accounted for these concerns by including a group of $\gamma 2^{f/f}$ mice injected with a control vector expressing only dTomato and treated with CNO (Fig. 4). Tone fear extinction responses in these animals were indistinguishable from those of both hM3Dq/dTomato-injected and naive $\gamma 2^{f/f}$ mice, indicating that CNO alone had no detectable effects in the absence of DREADD expression; this important point is now more clearly addressed in page 13, lines 264-266 and 275. Together, these findings validate the specificity and efficacy of our chemogenetic manipulation and address concerns about pharmacological confounds.

The authors attempted to rule out auditory deficits as a confounding factor for fear conditioning by measuring the startle responses. This test uses suprathreshold sound levels and thus does not assess auditory function, it just measures the startle response (of course, if the mice are deaf, they could detect a problem). There are standard methods to measure auditory function in mice.

We acknowledge the reviewer's point that the startle response test primarily measures reflexive responses to supra-threshold sound levels rather than providing a comprehensive assessment of auditory function. However, we consider that a full auditory evaluation is beyond the scope of this study. Importantly, we tested startle responses using both lower and higher decibel levels than those used in the fear conditioning paradigm and observed no significant differences between groups. Therefore, while we cannot entirely rule out subtle auditory impairments, our results suggest that they are unlikely to account for the observed differences in fear conditioning. To take into account the reviewer's comment, we amended our conclusion on auditory sensitivity (page 8, line 152-158; Page 9, line 164-165) and added a citation in hidden hearing loss (Ji et al., 2024, Plos Biol; Reference #26).

It is unclear why they used the GCaMP3 floxed allele mice without performing calcium imaging experiments. Calcium imaging could have provided critical, real-time insights into the functional dynamics of OPCs and their interactions with PV interneurons.

We initially generated the transgenic mice with the *GCaMP3* floxed allele to perform calcium imaging experiments in oligodendroglia. However, our previous study showed no detectable changes on GABA-A receptor-dependent calcium signals in OPCs between control and $\gamma 2$ mice (Balía et al., 2017, *Glia* PMID: 28795438). Moreover, it has been demonstrated that calcium activity in myelin sheaths decreases with age in the postnatal cortex (Battefeld et al., 2019, *Cell Rep* PMID: 30605675), a finding we have recently confirmed in OPCs and OLs in the adult corpus callosum (unpublished data). Given these results, calcium imaging would probably not provide meaningful information into PV interneuron-OPC interactions in our experimental context. Instead, the current study focuses on the broader impact of these interactions at the neuronal network and behavioral levels, which we believe provides the most relevant understanding of the role of oligodendroglia in circuit dynamics. We have now clarified in the Methods section that *GCaMP3* was used as solely a reporter (page 35, Line 777).

Other points:

1 – Methods: Description of the experiments performed to measure startle response is lacking.

We apologize for omitting the description of the startle response in the Methods. We have now added this description in the Behavior section of the Methods section (page 28, line 607-612):

2 – The representative images in the Extended Data Figure 4 suggest a reduction in PV axon density. This should be tested.

As seen in the Supplementary Figure 4a, PV staining appears dense due to the presence of PV+ interneuron dendrites, somata, synapses and axons. To specifically assess PV+ axon density, we analyzed the image stacks acquired with a 93X objective on the SP8 Leica microscope. We applied a mask created with SMI-312+ axon labeling to isolate axonal regions and then quantified the intensity of PV+ pixels within this mask. While this method does not directly measure PV+ axon density, it provides an estimate of PV signal in axonal compartments. Our analysis did not reveal significant differences between control and $\gamma 2$ mice. These changes are now reported in new Supplementary Figure 6f-g, Results section (page 11, line 227-229) and Methods section (page 36, line 800-803).

3 – On page 10, lines 183/184, the text incorrectly states that CNO treatment was applied in controls. It was only administered to $\gamma 2f/f$ mice.

Thank you for bringing this mistake to our attention. We now corrected the text accordingly.